- Synergistic identification of hydrogeological parameters and pollution
- source information for groundwater point and areal source
- contamination based on machine learning surrogate-artificial
- hummingbird algorithm
- Chengming Luo<sup>1</sup>, Xihua Wang<sup>1,2\*</sup>, Y. Jun Xu<sup>3</sup>, Shunqing Jia<sup>1</sup>, Zejun Liu<sup>1</sup>, Boyang Mao<sup>1</sup>,
- Qinya Lv<sup>1</sup>, Xuming Ji<sup>1</sup>, Yanxin Rong<sup>1</sup>, Yan Dai<sup>1</sup>
- <sup>1</sup> College of Civil Engineering, Tongji University, 1239 Siping Road, Shanghai 200092,
- China

7

14

17

18

19

- <sup>2</sup> Department of Earth and Environmental Sciences, University of Waterloo, ON N2L
- 11 3GI, Canada
- <sup>3</sup> School of Renewable Natural Resources, Louisiana State University Agricultural
- 13 Center, Baton Rouge, Louisiana, USA
- Email: 21531@tongji.edu.cn (Xihua Wang)
- 16 Tel.: + 0086 0431 13089410676; Fax: + 0086 021 65986809

# **Graphical Abstract**

# Highlights

- A highly adaptable inversion framework is adapted to different groundwater pollution
- scenarios.

- Synergetic identification of source information, hydraulic conductivity and boundary
- condition in PSC.
- The artificial hummingbird algorithm is applied to solve the optimized model.

#### **Abstract**

Effectively remediating groundwater contamination relies on the precise determination of its sources. In recent years, a growing research focus has been placed on concurrently estimating hydrogeological characteristics and locating pollutant origins. However, the identification of precise synergistic identification of point and areal contamination sources of groundwater and combined hydrogeological parameters has not been effectively solved. This study developed an inversion framework that integrates machine learning surrogates with the artificial hummingbird algorithm (AHA). The surrogate models approximating the simulation system were constructed using both backpropagation neural networks (BPNN) and Kriging techniques. The AHA was then employed to solve the optimized model, and its performance was benchmarked against particle swarm optimization (PSO) and the sparrow search algorithm (SSA). The applicability of this inversion framework was assessed by application to point sources of contamination (PSC) and areal source contamination (ASC). The robustness of the framework was verified through application to scenarios with different noise levels. The results showed that surrogate model constructed by the BPNN method provided estimates that were closer to those of the simulation model in comparison to the kriging method, coefficient of determination  $(R^2)$  is 0.9994 and mean relative error (MARE) is 3.70% in PSC, and R<sup>2</sup> is 0.9989 and MARE is 4.48% in ASC. The performance of the AHA exceeded those of the PSO and the SSA. In PSC, MARE of the identification result is 1.58%; In ASC, MARE of the identification result is 2.03%, with the AHA able to rapidly and accurately identify the global optimum and improve the inversion

- efficiency. The proposed inversion framework was demonstrated to apply to both
- groundwater PSC and ASC problems with strong robustness, providing a reliable basis
- for groundwater pollution remediation and management.
- **Keywords:** Groundwater contamination identification; Synergistic identification; Point
- and areal sources contamination; Surrogate model; Artificial hummingbird algorithm

#### 1 Introduction

Groundwater pollution adversely affects human production and life (Wang et al., 2022; Liu et al., 2024). The remediation of groundwater contamination is important for ensuring human health and socioeconomic development. However, groundwater contamination is difficult to detect and treat due to its hidden nature, thereby complicating the assessment of groundwater pollution risk and contamination liability (Li et al., 2021). Remediation requires the identification of sources of groundwater contamination (location, number, release history, etc.) and hydrogeological conditions (Maliva et al., 2015; Daranond et al., 2020; Pan et al., 2022b; Medici et al., 2024). However, directly obtaining this information can pose a challenge, with a proven method being the identification of groundwater contamination by inversion of limited observational data. Inversion of groundwater aquifer hydrogeologic parameters and pollution source information is a widely studied topic. In past studies on groundwater contamination identification (GCI), many researchers have focused on the separate identification of hydrogeological parameters or pollution source information. For example, Singh and Datta (2007) utilized backpropagation-based artificial neural network techniques specifically for the identification of groundwater pollution sources. Similarly, Mahar and Datta (2000) employed a nonlinear optimization model to identify the location, duration, and magnitude of the contamination source. Liu et al. (2022) inverted hydrogeological parameters through a simulation-optimization approach, while Wang et al. (2024a) combined three different inversion algorithms and a kriging surrogate

model to invert hydraulic conductivity. While simplifying the problem, these methods allow researchers to focus on specific aspects. However, although the individual identification method can be effective in some cases, it often overlooks the interconnectivity between hydrogeological parameters and pollution sources.

Currently, the simultaneous identification of hydrogeological parameters and pollution source information is gaining increasing attention in research. Researchers have employed various advanced technologies to achieve this goal. Wang et al. (2021) utilized a parallelized heuristic algorithm to concurrently determine both aquifer characteristics and the groundwater pollution sources. Pan et al. (2021) integrated a Bayesian-regularized deep neural network surrogate to jointly infer pollution source details and hydraulic conductivity. Hou et al. (2021) integrated homotopy-based inverse optimization theory with a multi-kernel extreme learning machine to finish the co-identification of contamination sources and aquifer parameters. Luo et al. (2023) leveraged machine learning techniques to establish an inverse relationship between model outputs and inputs, enabling fast and simultaneous retrieval of pollution source attributes and hydrogeological properties. Although these methods have advanced the field, improving recognition accuracy remains a major challenge in the simultaneous identification process.

The simulation-optimization method has been widely applied in GCI research because of its robust mathematical foundation (Mirghani et al., 2009) and its ability to identify multiple variables simultaneously. To enhance both identification accuracy and efficiency using simulation-optimization, two key approaches are employed: one is to

optimize the model solution method for better performance, and the other is to construct a surrogate model with high approximation accuracy. Optimizing the model solution method is essential. Since heuristic optimization algorithms are more capable of identifying global optima, many have been applied to GCI. Mirghani et al. (2012) implemented a genetic algorithm within optimization to identify sources of contamination. Jiang et al. (2013) combined a harmony search algorithm with a contamination transport simulation model to characterize contamination sources. Additional methods, such as simulated annealing (Rao, 2006; Yeh et al., 2007; Jha and Datta, 2013) and sparrow search algorithms (SSA) (Pan et al., 2022b), have also been applied to GCI. However, increasing dimensionality and complexity in GCI problems make it difficult for many optimization algorithms to efficiently search for global optima. Constructing high-accuracy surrogate models is another crucial strategy. Surrogate models can significantly reduce computation time and improve inversion efficiency. Among these models, the widely used kriging (Chugh et al., 2018; Zhang et al., 2019; Jiang et al., 2020) and backpropagation neural network (BPNN) (Sargolzaei et al., 2012; Zhang et al., 2021; Wang et al., 2024b) methods offer high flexibility and strong nonlinear fitting capabilities. Despite these advances, previous studies have overly focused on point source contamination (PSC) or areal source contamination (ASC) scenarios in isolation. However, the identification of precise synergistic identification of PSC and ASC of groundwater and combined hydrogeological parameters has not been effectively solved.

Based on the above problems, this paper proposes an inversion framework

algorithm (AHA) using the simulation-optimization method (Fig. 1). Both BPNN and kriging were utilized to develop surrogate models for the simulation model. AHA was introduced to solve the optimization model, with its solution results compared against those of PSO and SSA. The applicability of this inversion framework was evaluated through its application to both PSC and ASC scenarios. The objectives of this study were: (1) Develop a flexible groundwater pollution inversion scheme that can reliably invert parameters under various groundwater pollution scenarios; (2) Adopt an integrated parameter identification strategy to achieve the simultaneous identification of multiple variables, including pollutant release characteristics and hydrogeological parameters; (3) Design an optimization-based surrogate modeling method combining meta-heuristic search algorithms with neural network surrogate models to efficiently explore the solution space and reduce the risk of getting stuck in local optima during inversion calculations; (4) Evaluate the performance of the proposed scheme under various noise intensities and pollution patterns to validate its robustness and application potential in groundwater pollution inversion problems. The main innovations are as follows: (1) This study constructed an adaptive inversion framework that maintains high robustness in both PSC and ASC. (2) In PSC case, synergistic identification of source information, hydraulic conductivity, and boundary conditions. (3) Apply the AHA optimization model to solve the inverse problem of groundwater pollution to obtain the global optimal solution of the inverse problem and

integrating a machine learning surrogate model with the artificial hummingbird

further improve the inversion accuracy. The good compatibility between AHA and the

BPNN surrogate model ensures the robustness and stability of the inversion process.

## 2. Methodology

#### 2.1. Simulation model

- In this study, the numerical groundwater simulation framework comprised both a flow
- component and a solute transport module. The fundamental two-dimensional (2D)
- partial differential equation governing groundwater flow is formulated as follows:

$$\frac{\partial}{\partial x_i} (K_{ij} (H - z) \frac{\partial H}{\partial x_j}) + W = \mu \frac{\partial H}{\partial t} (x, y) \in S \ i, j \in 1, 2 \ t \ge 0$$
 (1)

- where  $K_{ij}$  is hydraulic conductivity, W is the volumetric flux per unit volume,  $\mu$  is the
- specific yield, H is the water level elevation, z is the elevation of the aquifer floor, and
- S is the boundary of the spatial domain.

$$\frac{\partial C}{\partial t} = \frac{\partial}{\partial x_i} (D_{ij} \frac{\partial C}{\partial x_i}) - \frac{\partial}{\partial x_i} (u_i C) + \frac{R}{n_e}$$
 (2)

$$u_{i} = \frac{K_{ij}}{n_{e}} \frac{\partial H}{\partial x_{i}}$$
 (3)

where C denotes the contaminant concentration in groundwater, t is the temporal

variable,  $u_i$  indicates the average flow velocity, R accounts for source and sink

contributions,  $D_{ij}$  refers to the hydrodynamic dispersion tensor, and  $n_e$  represents the

effective porosity of the medium. We used the MODFLOW-2005 (Harbaugh., 2005)

and MT3DMS (Zheng et al., 2012) numerical models to obtain numerical solutions for

groundwater flow and solute transport equations. (Asher et al., 2015).

#### 2.2. Kriging method

Kriging was employed to develop the underlying framework of the approach by

capturing both the correlation and stochastic variability of variables within a confined

spatial domain, thereby enabling the estimation of optimal regional values. The association between input and output variables is described through a regression-based expression as shown below (Zhao et al., 2022a):

$$y(x) = \sum_{i=1}^{k} \beta_{1i} f_i(x) + z(x)$$
 (4)

- where  $\hat{y}(x)$  is the estimated value of pollutant concentration y(x),  $f_i(x)(i =$
- $1, \dots, k$ ) is the basis function of the known regression model, and z(x) is the random
- part.
- The following equations were satisfied:

$$\begin{cases} E(z(x)) = 0 \\ D(z(x)) = \sigma^2 \\ \operatorname{cov}[z(x_i), z(x_j)] = \sigma^2 R(x_i, x_j) \end{cases}$$
 (5)

where  $R(x_i, x_j)$  is the correlation function between the sampled point  $x_i$  and  $x_j$ .

$$(i = 1, 2, \dots, m; j = 1, 2, \dots, m)$$

The Gaussian model is commonly used:

$$R(x_i, x_j) = \exp\left(-\sum_{k=1}^m \theta_k \left| x_{k_i} - x_{k_j} \right|^2\right)$$
 (6)

where  $\theta_k$  is a coefficient to be determined, which can be obtained by calculation.

## 2.3. The BPNN method

- A typical back-propagation neural network (BPNN) is composed of three fundamental components (Fig. 2): (1) an input layer, (2) the hidden layers, and (3) an output layer. The computation process proceeds in two main phases: forward propagation and backward propagation (Chen et al., 2010; Zhang et al., 2018).
- 1) During forward propagation, data are introduced into the network via the input

- layer, and subsequently processed through successive layers to yield the final output.
- BPNNs frequently employ a nonlinear sigmoid activation function:

$$f(x) = \frac{1}{1 + e^{-x}} \tag{7}$$

The calculation of the forward transmission output layer is:

$$I_{j} = \sum_{i=1}^{j} w_{ij} o_{i} + b \qquad o_{j} = f(I_{j}) = \frac{1}{1 + e^{I_{j}}}$$
 (8)

- where  $O_i$  represents the output of neuron i,  $O_j$  is the output of neuron j, b is the bias
- term, and  $W_{ij}$  is the weight of the connection between neuron i and neuron j.
- 2) Backward propagation involves the random assignment of the weight of the first
- positive feedback process within the output layer. The adjustment of the parameters of
- the entire network is required. Network adjustment is performed by minimizing the
- discrepancy between the predicted output and the target category in the output layer.
- Specifically, for the output layer:

$$E_{j} = O_{j}(1 - O_{j})(T_{j} - O_{j})$$
 (9)

- where  $E_j$  represents the error value at the jth node and  $T_j$  denotes the corresponding
- output. The hidden layer's output is determined by summing the weighted contributions
- from the errors of the lower nodes:

$$E_{i} = O_{i}(1 - O_{i}) \sum_{k} E_{k} W_{ik}$$
 (10)

- where  $E_k$  is the error gradient for the subsequent node k and  $W_{jk}$  is the weight connecting
- node j to t node k. Following error calculation, the weight is adjusted according to the
- error gradient:

$$\Delta W_{ij} = \eta E_j O_i$$

$$W_{ij} = W_{ij} + \Delta W_{ij}$$
(11)

where  $\eta$  is the learning rate. In Case 1, the BPNN architecture was configured as 19-30-

45, and in Case 2 as 15-20-50. The number of neurons in each layer was empirically optimized using grid search combined with cross-validation to minimize the root mean square error (RMSE) and effectively prevent overfitting. The sigmoid function was employed as the activation function, and the network was trained using the Bayesian Regularization algorithm. The maximum number of training iterations was set to 1000, and the learning rate was set to 0.01.

## 2.4 Artificial Hummingbird Algorithm (AHA)

The AHA consists of three main elements: food sources, hummingbirds, and the visit table. Hummingbirds typically assess food sources based on factors such as nectar quality, individual flower nectar content, and replenishment rates. For simplicity, it can be assumed that all food sources share the same flower type and number. Hummingbirds within a population can exchange information, be assigned to specific food sources, track nectar replenishment rates, and record the duration each food source remains unvisited. The visit table records the time since a hummingbird last visited a food source, and is used to assign visit levels; hummingbirds can harvest more nectar by first accessing food sources with higher access levels, following which food sources with the highest nectar replenishment rate are chosen (Zhao et al., 2022b). The AHA is algorithmically described below.

#### 226 (1) Initialization

Firstly, n hummingbirds are randomly placed on n food sources:

$$x_i = Low + r \cdot (Up - Low) \quad i = 1, ..., n$$
 (12)

The access table for the food source is then initialized:

$$VT_{i,j} = \begin{cases} 0 & if \quad i \neq j \\ \text{null} & i = j \end{cases} \quad i = 1, ..., n; \ j = 1, ..., n$$
 (13)

- where Low and Up are the lower and upper boundaries for a d-dimensional problem
- respectively, r represents a random vector of [0,1], and  $x_i$  is the position of the ith food
- source. For i = j,  $VT_{i,j} = null$  indicates the sourcing of food from a specific source.
- For  $i \neq j$ ,  $VT_{i,j} = 0$  indicates that the ith humming bird has just visited the jth food
- source in the current iteration.
- (2) Guided foraging

229

Hummingbirds identify food sources in two steps: (1) identifying the food source with the highest access level; (2) selecting the food source with the highest nectar replenishment rate. After identifying the target food source, the hummingbird can fly to the target source to feed. During foraging, direction switching vectors used to control the availability of one or more directions in the D-dimensional space are introduced to model three flight skills: omnidirectional, diagonal, and axial flight. These flight models can be extended to the d-D space, and the mathematical model of axial flight is:

$$D^{(i)} = \begin{cases} 1 & if \quad i = randi([1,d]) \\ 0 & else \end{cases} \quad i = 1, \dots, d$$
 (14)

Diagonal flight is defined as:

$$D^{(i)} = \begin{cases} 1 & if \quad i = P(j), j \in [1, k] \\ P = randperm(k), k \in [2, [r_1 \cdot (d-2)] + 1] & i = 1, ..., d \\ 0 & else \end{cases}$$
 (15)

Omnidirectional flight is defined as:

$$D^{(i)} = 1 i = 1, ..., d (16)$$

- where randi([1,d]) is a randomly generated integer from 1 to d, randperm(k)
- creates a random permutation of integers from 1 to k, and  $r_1$  is a random number in the
- range of 0 to 1.
- Hummingbirds can access and obtain target food sources through these flight abilities.
- New food sources identified during the search are recorded along with previously
- identified food sources. The guided foraging behavior and candidate food sources can
- be represented as:

$$v_{i}(t+1) = x_{i,tar}(t) + a \cdot D \cdot (x_{i}(t) - x_{i,tar}(t))$$
 (17)

$$257 a \sim N(0,1) (18)$$

- where  $x_{i,tar}(t)$  is the location of the food source that the *i*th humming bird plans to
- visit,  $x_i(t)$  represents the location of the *i*th food source at time t, and a is a leading
- factor obeying a normal distribution.
- The location of the *i*th food source is updated as:

- where  $f(\cdot)$  represents the function fitness value. The formula for updating the location
- can contribute to the preferential selection of food sources with a high nectar supply
- rate.

## (3) Territorial foraging

Since the quality of food sources within a foraging area may vary, hummingbirds actively search within that area. The regional foraging strategies and candidate food sources of hummingbirds can be represented as:

$$v_{i}(t+1) = x_{i}(t) + b \cdot D \cdot x_{i}(t)$$
 (20)

$$271 b \sim N(0,1) (21)$$

where *b* is a territorial factor obeying a normal distribution. Eq. (20) allows different hummingbirds to use their specific flight skills to identify new food sources near the target source.

#### (4) Migration foraging

Migration coefficients are defined in the AHA algorithm to prevent the generation of local optimums. The exceedance of the number of iterations of the set migration coefficient results in the hummingbird located in the worst food source repeating a search for a new food source across the entire search range and the subsequent updating of the visit table.

$$x_{war}(t+1) = Low + r \cdot (Up - Low)$$
 (22)

where  $x_{wor}$  is the food source with the worst nectar supply rate. The migration coefficient relative to population size can be defined as.

$$284 M = 2n (23)$$

#### 3. Case studies

The present study designed a groundwater PSC case study and an ASC case study to verify the applicability of the proposed GCI framework. Since the present study established two hypothetical examples, a set of variables to be identified and background variables for input into the groundwater contamination simulation model were established for each example for forward computation. The pollutant concentrations monitored at wells were used as observed data. The robustness of the inversion framework was verified by adding random noise to the observed data, expressed as:

$$\alpha_1 = \alpha(1 + l \cdot \text{rand}), \ l = 0.5\%, 1\% \text{ and } 2\%$$
 (24)

where  $\alpha$  represents the observation data,  $\alpha_1$  indicates observation data with added noise, l is the max disturbance range, and rand is a random number between -1 and 1.

## 3.1 Case study 1: groundwater PSC

The study area is 2,500 m and 1,400 m from east to west and north to south, respectively, with topography decreasing from west to east and groundwater flow from northwest to southeast. The study area contains a heterogeneous isotropic aquifer, and the present study focused on a layer of diving aquifer with a thickness of 10 m (Table 1). The aquifer comprises unconsolidated sediments, primarily well-sorted coarse sand and gravel. Groundwater flow was represented as 2D steady flow, and the study area was divided into three areas according to differences in hydraulic conductivities. Since the northern and southern parts of the study area are very weakly permeable formations, they were generalized in the present study as no-flow boundaries. Rivers formed the

boundaries of the western and eastern parts, and were generalized as specific head boundaries (Fig. 3).

In this case study, the variables to be identified fell into three main categories: (1) head values at the specific head boundaries. including  $H_1$  and  $H_2$ ; (2) hydraulic conductivities for each part of the study area, including  $K_1$ ,  $K_2$ , and  $K_3$ ; (3) the intensities of the release of pollutants from the two sources during the release periods:  $S = S_a T_b$ ; a = 1, 2; and b = 1, 2, 3, 4, 5 (Table S1).  $S_a T_b$  represents the intensity of pollution source a during the bth stress period; this case study had a study period of 10 years (Table 1, Fig. 4), with both sources only releasing pollutants in the first five years (Table S2). Five wells were established to monitor the concentrations of groundwater contaminants once a year. The study area was spatially discretized into 50 m × 50 m grids (Table 1).

#### 3.2 Case study 2: groundwater ASC

The present study selected the hypothetical case study used by Pan et al. (2022a) as a case study. The site has an area of 5 km<sup>2</sup>, with a length of 2.5 km and width of 2 km from east to west and south to north, respectively. Groundwater flows from northwest to southeast. The study area was conceptualized as a heterogeneous isotropic aquifer and the current study focused on a diving aquifer, in which flow was represented as 2D steady flow. The study area's aquifers were categorized into four zones based on hydraulic conductivity, labeled  $K_1$  to  $K_4$ . The western and eastern river boundaries were modeled as specified head boundaries, while the northern and southern regions, characterized by low permeability granite, were treated as no-flow boundaries (Fig. 5, Table 2). The aquifer comprises unconsolidated sediments, primarily well-sorted coarse

sand and gravel.

Within this case study, the variables to be identified fell into two categories: (1) hydraulic conductivities of each part of the study area, including  $K_1$  to  $K_4$ ; (2) the intensities of pollutants released by three areal sources of contamination:  $S = S_a T_b$ ; a = 1, 2, 3; and b = 1, 2, 3, 4, 5 (Table S3).  $S_a T_b$  indicates the intensity of pollution source a during the bth stress period. A total of nine monitoring wells were established to monitor the concentrations of groundwater contaminants once a year (Fig. 6). The study area was spatially discretized as 20 m × 20 m grids (Table 2).

#### 4. Model construction

### 4.1 Establishment of surrogate models

The present study established two case studies: the PSC and the ASC. The variables to be identified for the PSC case study included three categories with 15 dimensions, whereas those to be identified for the ASC case study included two categories with 19 dimensions. The present study used the Latin hypercube method to sample within the feasible domain of the variables to be identified. This sampling process was implemented in MATLAB. Sample groups for the PSC and ASC case studies totaled 390 and 490, respectively, and the input sample dataset was generated by random combination.

The parameters obtained from the above sampling were input into the groundwater simulation model. The simulation model was then run to obtain the pollutant concentrations at the 390 and 490 monitoring groups in the PSC and ASC case studies,

respectively. These simulated pollutant concentrations were used as the output sample

dataset, and the output sample dataset was combined with the input sample dataset to form the input-output sample dataset. The kriging and BPNN methods were used to establish the surrogate models of the simulation model. The first 350 and 440 groups of the PSC and ASC case input-output sample datasets, respectively, were used as training samples in each case study to construct surrogate models, while the remaining 40 and 50 groups were used as test samples to evaluate the accuracy of the surrogate models.

The present study applied the coefficient of determination (R<sup>2</sup>), the mean absolute relative error (MARE), and the root mean square error (RMSE) to assess the accuracy of the fit of the estimations of the surrogate models to the output of the simulation model.

1)  $R^2$ : The closer  $R^2$  to 1, the more accurate the surrogate model is.

$$R^{2} = 1 - \frac{\sum_{i=1}^{n} (y_{i} - \hat{y}_{i})^{2}}{\sum_{i=1}^{n} (y_{i} - \overline{y}_{i})^{2}}$$
 (25)

2) MARE: The average deviation between the outputs of the surrogate model and the outputs of the simulation model.

$$MARE = \frac{\sum_{i=1}^{n} \left| \frac{y_i - \hat{y}_i}{y_i} \right|}{n}$$
 (26)

3) RMSE: The value of the RMSE is inversely proportional to the fitting accuracy of the surrogate model.

$$RMSE = \sqrt{\frac{\sum_{i=1}^{n} (y_i - \hat{y}_i)^2}{n}}$$
 (27)

where  $\overline{y}_i$  is the average true value, n is the number of samples,  $\hat{y}_i$  is the output of the surrogate model,  $y_i$  is the true value of the variable to be identified.

### 4.2 Establishment of the optimization models

This study employed the CGI through the S-O method, which consists of two main components: a groundwater contaminant transport simulation model and an optimization model aimed at minimizing the least squares error between the simulated and true values. To reduce the computational burden caused by repeated simulation calls, a surrogate model was used in place of the simulation model. While the same objective function was applied in both case studies, there were minor variations in the decision variables and constraints. The decision variables chosen for case study 1 included the boundary head values, the hydraulic conductivities of the site, and the release history of the contaminant source; those for case study 2 included the hydraulic conductivities of the site and the release history of the contaminant source. The constraint conditions were influenced by the decision variables. The optimization was expressed as:

$$z = \min \sum_{m=1}^{n} (C_m - \hat{C}_m)^2$$

$$Case 1:s.t \begin{cases} C = f(H, K, s) \\ C_L \le C \le C_U \\ s_l \le s \le s_u \end{cases}$$

$$Case 2:s.t \begin{cases} C = f(K, s) \\ C_L \le C \le C_U \\ s_l \le s \le s_u \end{cases}$$

$$(28)$$

where z is the objective function,  $C_m$  is the monitored pollutant concentration in the mth monitoring well,  $\hat{C}_m$  is the simulated pollutant concentration in the mth monitoring well, C is the pollutant concentration, H is the head value at the boundary, S is the pollution source intensity, K represents the hydraulic conductivities of the site,

 $C_L$  and  $C_U$  are the upper and lower bound values of pollutant concentration, respectively, and  $s_l$  and  $s_u$  are the upper and lower bound values of pollution source intensity, respectively.

The AHA was used to identify the optimal combination of parameters according to the objective function through multiple iterative calculations, with this parameter set adopted as the result of inversion. The numbers of hummingbird populations and iterations were set to 500 and 1,000, respectively.

#### 5. Results

## 5.1 Surrogate models

The surrogate model for case study 1 using the kriging method achieved an R<sup>2</sup> of 0.9942, MARE of 13.43%, and RMSE of 11.8262 (Table 3), while the BPNN method produced values of 0.9994, 3.70%, and 3.6526, respectively (Table 3). Similarly, for case study 2, the kriging method yielded an R<sup>2</sup> of 0.9837, MARE of 9.98%, and RMSE of 37.7547, whereas the BPNN method provided corresponding values of 0.9989, 3.70%, and 3.6526 (Table 3). The BPNN method demonstrated superior goodness-of-fit statistics compared to the kriging method in both case studies. While the simulation model required 50 hours for 1,000 iterations, the BPNN surrogate model completed the same number of iterations in 67 seconds, significantly reducing the computation time.

### **5.2 Optimization algorithms**

The BPNN surrogate model was embedded into the optimization model to optimize the parameter combination according to the objective function. This study employed AHA within the optimization process and compared its performance against SSA and PSO

under the same population size and number of iterations. In the optimization of case study 1, PSO failed to converge after reaching the maximum number of iterations, while AHA and SSA converged after 120 and 350 iterations, respectively (Fig. 7a). For case study 2, both PSO and SSA failed to converge within the maximum number of iterations,

Given the results from case study 1, where both AHA and SSA converged, the subsequent analysis focused on these two algorithms. AHA achieved an optimal search value closer to the true value and reached the global optimum, while SSA settled at a local optimum (Fig. 8). These results demonstrate that AHA not only converged faster than SSA but also identified the global optimum, thereby improving the accuracy and

#### 5.3 Inversion results and robustness assessment

efficiency of GCI.

whereas AHA converged after 150 iterations (Fig. 7b).

The BPNN-AHA inversion framework developed in this study was applied to identify groundwater PSC and ASC and obtain inversion values. To verify the framework's robustness and reliability, random noise levels of 0.5%, 1%, and 2% were added to the observed data. The average relative errors under each noise level were recorded (Table 4, Table 5). The highest inversion accuracy was achieved in the noise-free case for both case study 1 and case study 2, with average relative errors of 1.58% and 2.03%, respectively (Table S4). At a 0.5% noise level, the average relative errors for case study 1 and case study 2 were 1.71% and 2.3%. At 1% noise, they were 2.03% and 2.33%, while at 2% noise, they increased to 2.55% and 3.52%, respectively. Although noise impacted the inversion accuracy, the framework maintained high performance, with the

average relative errors for both case studies remaining below 5% (Fig. 9). These results confirm the strong robustness and stability of the proposed inversion framework.

There are significant differences in sensitivity to noise among different parameter categories. Hydraulic conductivity: These parameters showed low sensitivity to noise, with relative errors remaining below 3% in all scenarios for both PSC and ASC cases. Their errors increased gradually with noise but remained stable, indicating strong robustness. Boundary head values (PSC case only): These parameters also exhibited excellent noise resistance, with relative errors consistently below 1% even at 2% noise level. Source release intensities: This group showed the highest sensitivity to noise. At a 2% noise level, some source parameters (e.g.,  $S_1T_1$  in PSC,  $S_1T_3$ ,  $S_1T_4$ ,  $S_3T_2$ ,  $S_3T_3$ ,  $S_3T_5$  in ASC) had relative errors exceeding 6%–10%, reflecting their higher inversion uncertainty under noisy conditions.

#### **6 Discussion**

#### 6.1 Analysis of surrogate models

The results of this study show that the proposed BPNN–AHA framework achieves high accuracy, strong robustness, and efficient convergence in GCI tasks, performing consistently well in both PSC and ASC scenarios, even under varying noise levels. In the PSC and ASC cases analyzed here, the R² values reached 0.9994 and 0.9989, and the MARE values were 3.70% and 4.48%, respectively, demonstrating the model's excellent capability to approximate the input–output relationships of the simulation model. The BPNN surrogate model, with its simple structure, high flexibility, and broad adaptability, effectively balances accuracy and generalizability—characteristics that are

essential for practical inversion applications. Compared to other surrogate modeling approaches reported in recent GCI research—such as long short-term memory neural networks (Li et al., 2021), light gradient boosting machines (Pan et al., 2023), and deep residual networks (Xu et al., 2024b)—the proposed framework leverages the adaptability of BPNN together with the global search and adaptive convergence mechanisms of the artificial hummingbird algorithm to deliver consistently accurate and stable inversion results. In this paper, the ASC is drawn from Pan et al. (2022a), which had been widely validated in other studies. For example, Li et al. (2023) used the same case to validate an inversion method, applying a multilayer perceptron model to the simulation, achieving the R<sup>2</sup> of 0.9999 and the MARE of 2.85%. Similarly, Xu et al. (2024a) employed automatic machine learning methods for surrogate model construction, achieving the R<sup>2</sup> of 0.9754 and the MARE of 4.154%. Compared to the surrogate models developed by these researchers, the BPNN model constructed in this study also demonstrates excellent approximation accuracy, further validating the advantages of the proposed method. In summary, the proposed BPNN surrogate model has practical advantages in tasks related to GCI, thereby enhancing its applicability. Due to its relatively simple architecture and low computational requirements, the BPNN model can be trained and updated efficiently even under limited computational resources. Additionally, the model demonstrates strong generalization capabilities in both PSC and ASC scenarios, indicating that it is not specific to a particular case. This adaptability is crucial for practical groundwater inversion problems, as data availability and system complexity often vary significantly across different locations. These

characteristics highlight the comprehensive advantages of the BPNN model in terms of accuracy, efficiency, and flexibility, making it a reliable and practical choice for surrogate modeling in groundwater simulation.

## 6.2 Analysis of optimization algorithms

This paper compares the AHA with PSO and SSA under the same preconditions and finds that AHA offers clear advantages in both convergence speed and global optimization capability. Based on these results, AHA was chosen to solve the optimization model, and its adaptability was further verified in two different cases. In the field of optimization algorithms, the "no free lunch principle" (Zhao et al., 2022b) emphasizes that no single algorithm performs well across all optimization problems. When addressing real-world problems, it is essential to understand the nature of the problem thoroughly before selecting the appropriate optimization algorithm. This principle encourages researchers to develop new and more effective algorithms from different perspectives, providing more options for optimization problem researchers. This insight also applies to groundwater pollution traceability. Given the diverse nature of pollution traceability problems, it is challenging for any single optimization algorithm to be universally applicable. As research deepens, these problems tend to become more high-dimensional and nonlinear, necessitating the exploration of algorithms with stronger global optimization capabilities and higher search efficiency. Additionally, it is important to consider alternative uses of optimization methods. One promising approach involves using optimization techniques to improve machine learning models by identifying optimal parameters (hyperparameters) during training,

which can significantly enhance model accuracy (Jia et al., 2024).

## **6.3 Inversion analysis**

Previous studies related to GCI employed a variety of methods to conduct either single or simultaneous inversion characterization of pollution sources and to identify hydrogeological parameters of the model. Li et al. (2022) identified the number, location, and release history of pollution sources, while Li et al. (2008) focused on determining the hydraulic conductivities of a study site. Bai et al. (2022) utilized inversion techniques to simultaneously characterize pollution sources and identify the hydraulic conductivities within their simulation models. While some studies have applied inversion to the boundary conditions of the simulation model (Jiao et al., 2019), fewer studies have simultaneously characterized pollution sources and identified both hydrogeological parameters and boundary conditions of the model. Source information, model hydrogeological parameters, and boundary conditions are all critical components of groundwater contamination simulation models. Inaccuracies in any of these components can affect the overall results of inversion, making it essential to identify all components simultaneously. Therefore, in the PSC case of this study, the release history of the pollutant source, the hydraulic conductivity of the model, and the specific head boundary values were simultaneously identified. This simultaneous identification of multiple key parameters enhances the reliability and effectiveness of decision support systems. In addition to the methods applied in this study, data assimilation methods are also widely used in the field of groundwater pollution inversion. They can combine

observational data with numerical models to improve state estimation and parameter inversion (Zafarmomen et al., 2024). Many researchers have successfully applied data assimilation methods to the iterative optimization of pollutant transport states and related parameters, significantly improving inversion accuracy and reducing prediction uncertainty. For example, Pan et al. (2022a) proposed a refined particle filter with a deep learning method surrogate as an inverse framework for groundwater pollution source estimation. This framework was evaluated under different levels of observational error through estimation tasks for point source pollution cases and nonpoint source pollution cases. Wang et al. (2023) utilized an improved particle filter method for groundwater pollution source identification. Zhang et al. (2024) used an iterative local updating ensemble smoother method to simultaneously identify pollution source information and hydraulic conductivity fields. However, both the method proposed in this study and data assimilation methods have their own advantages and disadvantages. The method proposed in this study possesses strong fine-grained search capabilities but its performance is highly dependent on the selection of initial points. Data assimilation methods can integrate multi-source data, significantly improving the spatio-temporal consistency of inversion results; however, their fine-grained search capabilities are somewhat limited. Future research could explore combining the realtime updating capabilities of data assimilation with the adaptability and optimization efficiency of the framework proposed in this study to further enhance the adaptability and performance of groundwater pollution inversion.

One of the main methodological motivations of this study is the integration of the

BPNN surrogate model with the AHA for GCI. This choice is grounded both in the inherent characteristics of GCI problems and in the complementary mechanisms of the two methods. GCI is a typical high-dimensional, nonlinear, and ill-posed inverse problem. The mapping from observed contaminant concentrations to source characteristics and hydrogeological parameters is often multimodal and nonconvex. In such cases, surrogate models such as BPNN can provide a fast and flexible approximation to computationally demanding groundwater simulations, but their use inevitably introduces approximation errors into the inversion objective function. These errors may create local irregularities in the objective function landscape, which can mislead optimizers and cause premature convergence—particularly when the optimization algorithm lacks a mechanism to balance exploration and exploitation adaptively. AHA offers notable advantages in addressing these issues. Its bio-inspired mode-switching strategy alternates dynamically between diversified search and focused search. In the early stages of optimization, the broad and varied exploration capability helps to survey the global search space and reduces the risk of becoming trapped in spurious local optima caused by surrogate-induced noise. As the search proceeds, the algorithm adaptively shifts toward more intensive exploitation, concentrating computational effort on promising regions and thereby accelerating convergence. This dynamic adjustment is particularly important in GCI problems, where the optimal parameter region is often narrow and embedded within a complex and noisy search space. In addition, AHA's adaptive update mechanism adjusts search trajectories in response to population feedback, effectively mitigating the influence of local

fluctuations in the surrogate-predicted objective function on the optimization process. This robustness to noisy or irregular fitness landscapes complements the BPNN's ability to generalize across diverse contamination scenarios. It is worth emphasizing that this integration is not a simple "algorithm replacement," but a targeted design choice based on the structural characteristics of the problem: BPNN provides broad adaptability to varying hydrogeological conditions, while AHA contributes resilience and fine-tuning capability when the optimization landscape is distorted by surrogate approximation errors. This synergy allows the proposed framework to maintain both high accuracy and strong robustness under different contamination scenarios and noise levels. More importantly, the underlying design principle—matching the characteristics of the surrogate model with the search dynamics of the optimization algorithm—has broader applicability to other environmental inversion problems.

#### **6.4 Limitations**

The overall inversion framework in this paper combines BPNN and AHA and is validated under different noise scenarios to account for the effect of noise in the observed data. The results indicate that the inversion framework demonstrates high robustness. However, a limitation of this paper is that noise is not addressed, and its presence can contaminate the observed data, further impacting the accuracy of GCI. Noise elimination methods could be applied to the observed data in future studies. Another major limitation is the generalization of the actual aquifer system. Groundwater systems are often complex, necessitating model simplifications through assumptions (e.g., homogeneity, isotropy) that may not reflect the actual geological

conditions, thereby affecting model accuracy. To address actual problems, the hydrogeological conditions of the study area should be thoroughly investigated, ensuring the model closely represents the actual situation, reducing error, improving model accuracy, and ultimately enhancing inversion accuracy. In terms of computational time and efficiency, by integrating the agent model, we can avoid repeatedly calling the numerical simulation model during the optimization process, thereby significantly improving computational efficiency. In our current implementation, thousands of optimization iterations can be completed in just a few minutes. However, as the complexity of the inversion problem increases, the number of required samples and the training time for the surrogate model will also increase significantly. Additionally, the current BPNN surrogate model is relatively lightweight, while deeper networks or ensemble-based surrogate models may require more computational resources. To address these issues, potential future solutions include parallel computing, adaptive sampling, and hybrid surrogate strategies that balance accuracy and efficiency.

#### 7 Conclusions

In this study, a BPNN-AHA inversion framework was developed to accurately and synergistically identify groundwater point and areal sources of contamination and combined hydrogeologic parameters. Among them, the BPNN surrogate model can well replace the simulation model, and the AHA had good global optimization capability and excellent solution accuracy. The robustness of the proposed methodology was verified by applying the inversion framework to scenarios with different noise

- levels. The conclusions of the present study are listed below:
- (1) The construction of a surrogate model to the simulation model satisfied the fitting
- accuracy requirement while also significantly reducing the computational time. The
- current study established BPNN and kriging surrogate models, with a comparison of
- the outputs of the models illustrating that the former obtained a higher fitting accuracy,
- with R<sup>2</sup> values of 0.9994 and 0.9989 for case 1 and case 2, respectively. Therefore, it
- can be applied to the inversion framework.

- (2) The present study applied AHA within the model optimization, with the results
- compared to those of PSO and SSA optimization. Compared to PSO and SSA, AHA
- rapidly reached convergence and identified the global optimum, the MAPE values for
- the inversion results of case 1 and case 2 were 1.58% and 2.03%, respectively.
- (3) The proposed inversion framework can realize the synergistic identification of PSC
- and ASC combined with hydrogeological parameters, which can ensure high
- identification accuracy, and the inversion framework has strong robustness under
- different noise levels. While individual identification simplifies the problem but may
- ignore correlations between parameters, synergistic identification improves the
- accuracy and consistency of identification by synchronizing the estimation of pollution
- sources and hydrogeological parameters. However, noise and parameter estimation
- of the inversion results. Therefore,
- uncertainty analysis needs to be further considered in subsequent studies. Overall, the
- BPNN-AHA inversion framework has excellent inversion performance and strong
- practicability, which can provide a reliable basis for groundwater pollution remediation

and management. For researchers working in groundwater contamination source identification, this study underscores that method selection should not be guided solely by algorithmic novelty, but should be informed by the inherent complexity of the problem and the compatibility between the research question and the chosen approach. In groundwater contamination inversion, selecting a highly compatible method can substantially improve efficiency, while leveraging and organically integrating the strengths of different methods can greatly enhance robustness. This concept is equally applicable to a broader range of complex environmental inversion problems, offering valuable insights and practical potential.

| 640 | Author's contributions                                                                 |
|-----|----------------------------------------------------------------------------------------|
| 641 | CL: Methodology, Formal analysis, Software, Conceptualization, Validation, Writing-    |
| 642 | original draft. XW: Supervision, Resources, Funding acquisition, Writing- review &     |
| 643 | editing. YX: Supervision, Writing-review & editing. SJ, ZL, & BM: Software, Formal     |
| 644 | analysis. QL, XJ, YR & YD: Methodology, Conceptualization.                             |
| 645 | Declaration of competing interest                                                      |
| 646 | The authors declared that they have no known competing financial interests or personal |
| 647 | relationships that could have appeared to influence the work reported in this paper.   |
| 648 | Acknowledgment                                                                         |
| 649 | This research was supported by the Fundamental Research Funds for the Central          |
| 650 | Universities (02002150257) and Overseas High-level Talents Program of Shanghai and     |
| 651 | Leading Talents (Overseas) Program of Shanghai.                                        |
| 652 | Data Availability                                                                      |
| 653 | The code used in this study can be found at https://doi.org/10.5281/zenodo.14568110.   |
| 654 |                                                                                        |

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

- Integration of DDPM and ILUES for Simultaneous Identification of Contaminant
- Source Parameters and Non-Gaussian Channelized Hydraulic Conductivity Field,

| 309 | Water              | Resour.             | Res.,             | 60,             | e2023WR036893,           |
|-----|--------------------|---------------------|-------------------|-----------------|--------------------------|
| 310 | https://doi.c      | org/10.1029/2023    | WR036893, 202     | 24.             |                          |
| 311 | Zhang, Y., Tang,   | J., Liao, R., Zha   | ng, M., Zhang,    | Y., Wang, X.,   | and Su, Z.: Application  |
| 312 | of an enhan        | ced BP neural ne    | twork model wi    | th water cycle  | algorithm on landslide   |
| 313 | prediction,        | Stochastic Enviro   | on. Res. Risk As  | ssess., 35, 127 | 3-1291, https://doi.org/ |
| 314 | 10.1007/s00        | 0477-020-01920-     | y, 2021.          |                 |                          |
| 315 | Zhao, Y., Fan, D   | o., Li, Y., and Yar | ng, F.: Applicati | on of machine   | e learning in predicting |
| 316 | the adsorpt        | ion capacity of o   | rganic compour    | nds onto biocl  | nar and resin, Environ.  |
| 317 | Res., 208, 1       | 12694, https://do   | i.org/10.1016/j.  | envres.2022.1   | 12694, 2022a.            |
| 318 | Zhao, W., Wang     | , L., and Mirjalil  | i, S.: Artificial | hummingbird     | algorithm: A new bio-    |
| 319 | inspired op        | timizer with its    | engineering ap    | plications, Co  | omput. Methods Appl.     |
| 320 | Mech. Eng.         | , 388, 114194, ht   | tps://doi.org/ 10 | .1016/j.cma.20  | 021.114194, 2022b.       |
| 321 | Zheng, C., Hill, N | M. C., Cao, G., an  | d Ma, R.: MT3D    | MS: MODEL       | USE, CALIBRATION         |
| 322 | AND VALI           | DATION, Trans.      | ASABE, 55, 15     | 49-1559, 2012   | 2.                       |
|     |                    |                     |                   |                 |                          |

Figure captions 824 Figure 1: General process used in the present study to construct the machine learning 825 826 surrogate model-artificial hummingbird algorithm framework. Figure 2: Structure of a back-propagation neural network (BPNN). 827 **Figure 3:** Schematic diagram of case study 1. 828 Figure 4: Distributions of concentrations of groundwater pollutants over different 829 periods: (a)–(j) represent 1–10 years. 830 Figure 5: Schematic diagram of case study 2. 831 Figure 6: Distributions of concentrations of groundwater pollutants over different 832 periods: (a) 1 year; (b) 2 years; (c) 3 years; (d) 4 years; (e) 5 years. 833 Figure 7: Convergence curves of the sparrow search algorithm (SSA), particle swarm 834 835 optimization (PSO), and artificial hummingbird algorithm (AHA) applied to case study. (a) case study 1; (b) case study 2. 836 Figure 8: Comparison between the true values and optimal values for the sparrow 837 search algorithm (SSA) and artificial hummingbird algorithm (AHA). 838 Figure 9: Comparison of relative errors for case studies 1 and 2 under different noise 839 levels. 840 841

842

**Figure 1.** General process used in the present study to construct the machine learning surrogate model-artificial hummingbird algorithm framework.

## Input layer Hidden layer Output layer

**Figure 2.** Structure of a back-propagation neural network (BPNN).

Figure 3. Schematic diagram of case study 1.

**Figure 4.** Distributions of concentrations of groundwater pollutants over different periods: (a)–(j) represent 1–10 years.

Figure 5. Schematic diagram of case study 2.

**Figure 6.** Distributions of concentrations of groundwater pollutants over different periods: (a) 1 year; (b) 2 years; (c) 3 years; (d) 4 years; (e) 5 years.

**Figure 7.** Convergence curves of the sparrow search algorithm (SSA), particle swarm optimization (PSO), and artificial hummingbird algorithm (AHA) applied to case study.

(a) case study 1; (b) case study 2.

**Figure 8.** Comparison between the true values and optimal values for the sparrow search algorithm (SSA) and artificial hummingbird algorithm (AHA).

**Figure 9.** Comparison of relative errors for case studies 1 and 2 under different noise levels.

Table 1 Fundamental values and ranges of aquifer parameters.

| Parameter                                            | Value or range |
|------------------------------------------------------|----------------|
| Hydraulic conductivity of zone 1, $K_1$ (m/d)        | (50,70)        |
| Hydraulic conductivity of zone 2, $K_2$ (m/d)        | (35,55)        |
| Hydraulic conductivity of zone 3, $K_3$ (m/d)        | (40,60)        |
| Specific yield of zone 1, $\mu_1$                    | 0.27           |
| Specific yield of zone 2, $\mu_2$                    | 0.22           |
| Specific yield of zone 3, $\mu_3$                    | 0.25           |
| Longitudinal dispersity of zone 1 (m)                | 40             |
| Longitudinal dispersity of zone 2 (m)                | 30             |
| Longitudinal dispersity of zone 3 (m)                | 35             |
| Grid spacing in X and Y direction (m)                | 50             |
| Recharge rate (m/d)                                  | 0.00042        |
| Initial concentration (mg/L)                         | 50             |
| Length of the stress period (y)                      | 10             |
| Aquifer thickness(m)                                 | 10             |
| Groundwater level at the western boundary, $H_1$ (m) | (18,20)        |
| Groundwater level at the eastern boundary, $H_2(m)$  | (15,17)        |

## Table 2 Fundamental values and ranges of aquifer parameters and pollution sources.

| Parameter                                                | Value or range |
|----------------------------------------------------------|----------------|
| Specific yield                                           | 0.24           |
| Transverse dispersity (m)                                | 9.8            |
| Longitudinal dispersity (m)                              | 40             |
| Aquifer thickness(m)                                     | 40             |
| Grid spacing in x-direction(m)                           | 20             |
| Grid spacing in y-direction(m)                           | 20             |
| Number of stress periods                                 | 5              |
| Hydraulic conductivity(m/d)                              | (30,50)        |
| Fluxes of contamination source during stress period(g/d) | (0,52)         |

Table 3 A comparison of the accuracies of the assessed surrogate models.

| Case  | Surrogate model | $\mathbb{R}^2$ | MARE   | RMSE    |
|-------|-----------------|----------------|--------|---------|
| C1    | Kriging         | 0.9942         | 13.43% | 11.8262 |
| Case1 | BPNN            | 0.9994         | 3.70%  | 3.6526  |
| C2    | Kriging         | 0.9837         | 9.98%  | 37.7547 |
| Case2 | BPNN            | 0.9989         | 4.48%  | 9.8488  |

Table 4 A comparison of inversion values under different noise levels for case study 1.

| Unknown   | Unknown True Inversion values under different noise levels |       |       |       |       |                |       |       |       |
|-----------|------------------------------------------------------------|-------|-------|-------|-------|----------------|-------|-------|-------|
| variables | value                                                      | 0     | 0.5%  | 1%    | 2%    | Relative error |       |       |       |
| $K_1$     | 60.37                                                      | 58.91 | 59.46 | 61.16 | 61.15 | 2.42%          | 1.50% | 1.31% | 1.29% |
| $K_2$     | 42.84                                                      | 42.12 | 41.73 | 41.72 | 42.18 | 1.67%          | 2.58% | 2.61% | 1.54% |
| $K_3$     | 50.17                                                      | 49.28 | 48.52 | 48.58 | 50.01 | 1.78%          | 3.29% | 3.17% | 0.31% |
| $H_1$     | 19.09                                                      | 19.10 | 19.04 | 19.06 | 19.27 | 0.06%          | 0.24% | 0.18% | 0.96% |
| $H_2$     | 16.11                                                      | 16.05 | 15.97 | 16.01 | 16.27 | 0.40%          | 0.87% | 0.64% | 0.97% |
| $S_1T_1$  | 34.25                                                      | 34.65 | 34.82 | 35.37 | 36.50 | 1.16%          | 1.66% | 3.26% | 6.57% |
| $S_1T_2$  | 57.07                                                      | 57.20 | 57.35 | 57.66 | 58.79 | 0.24%          | 0.49% | 1.04% | 3.01% |
| $S_1T_3$  | 5.80                                                       | 5.48  | 5.59  | 5.64  | 5.56  | 5.49%          | 3.63% | 2.78% | 4.19% |
| $S_1T_4$  | 31.76                                                      | 31.80 | 31.84 | 31.99 | 32.71 | 0.15%          | 0.25% | 0.74% | 3.00% |
| $S_1T_5$  | 18.14                                                      | 18.21 | 18.24 | 18.31 | 18.63 | 0.39%          | 0.55% | 0.96% | 2.73% |
| $S_2T_1$  | 82.07                                                      | 81.45 | 81.67 | 82.48 | 84.62 | 0.76%          | 0.50% | 0.49% | 3.10% |
| $S_2T_2$  | 22.18                                                      | 21.02 | 20.99 | 21.10 | 21.86 | 5.22%          | 5.37% | 4.87% | 1.44% |
| $S_2T_3$  | 74.35                                                      | 75.69 | 75.95 | 76.44 | 77.69 | 1.80%          | 2.15% | 2.81% | 4.49% |
| $S_2T_4$  | 4.92                                                       | 4.86  | 4.85  | 4.74  | 4.84  | 1.37%          | 1.48% | 3.76% | 1.78% |
| $S_2T_5$  | 15.84                                                      | 15.95 | 16.00 | 16.12 | 16.29 | 0.73%          | 1.06% | 1.81% | 2.86% |

Table 5 A comparison of inversion values under different noise levels for case study 2.

| Unknow         | True  | Inversion values under different noise levels |       |       |       |          |         |       |       |
|----------------|-------|-----------------------------------------------|-------|-------|-------|----------|---------|-------|-------|
| n<br>variables | value | 0                                             | 0.5%  | 1%    | 2%    | Relative | e error |       |       |
| $K_1$          | 45.93 | 44.94                                         | 45.44 | 45.07 | 46.01 | 2.15%    | 1.07%   | 1.87% | 0.17% |
| $K_2$          | 46.54 | 46.68                                         | 47.28 | 46.83 | 47.92 | 0.29%    | 1.59%   | 0.62% | 2.97% |
| $K_3$          | 32.11 | 32.08                                         | 31.91 | 32.05 | 31.73 | 0.08%    | 0.62%   | 0.20% | 1.19% |
| $K_4$          | 44.23 | 44.56                                         | 43.79 | 44.35 | 42.95 | 0.75%    | 0.98%   | 0.26% | 2.89% |
| $S_1T_1$       | 38.05 | 37.48                                         | 37.59 | 37.85 | 38.14 | 1.48%    | 1.22%   | 0.51% | 0.23% |
| $S_1T_2$       | 32.24 | 32.84                                         | 32.55 | 33.10 | 32.42 | 1.84%    | 0.95%   | 2.65% | 0.55% |
| $S_1T_3$       | 24.96 | 26.75                                         | 26.46 | 26.89 | 26.48 | 7.18%    | 6.01%   | 7.74% | 6.09% |
| $S_1T_4$       | 5.17  | 4.89                                          | 4.85  | 4.93  | 4.77  | 5.44%    | 6.33%   | 4.79% | 7.82% |
| $S_1T_5$       | 25.42 | 26.48                                         | 26.29 | 26.69 | 26.42 | 4.18%    | 3.43%   | 5.03% | 3.94% |
| $S_2T_1$       | 31.15 | 31.17                                         | 31.21 | 31.38 | 31.48 | 0.08%    | 0.19%   | 0.74% | 1.07% |
| $S_2T_2$       | 39.94 | 40.17                                         | 40.12 | 40.65 | 40.58 | 0.57%    | 0.43%   | 1.76% | 1.59% |
| $S_2T_3$       | 51.5  | 51.77                                         | 51.74 | 52.00 | 52.00 | 0.53%    | 0.47%   | 0.97% | 0.97% |
| $S_2T_4$       | 49.47 | 48.91                                         | 48.81 | 49.51 | 49.36 | 1.13%    | 1.33%   | 0.09% | 0.21% |
| $S_2T_5$       | 31.53 | 33.54                                         | 33.30 | 33.41 | 33.03 | 6.38%    | 5.61%   | 5.97% | 4.75% |
| $S_3T_1$       | 27.49 | 27.61                                         | 28.03 | 28.01 | 28.75 | 0.43%    | 1.96%   | 1.90% | 4.59% |
| $S_3T_2$       | 26.93 | 27.33                                         | 27.88 | 27.68 | 28.80 | 1.47%    | 3.52%   | 2.76% | 6.95% |
| $S_3T_3$       | 5.95  | 5.97                                          | 6.14  | 6.11  | 6.38  | 0.27%    | 3.15%   | 2.66% | 7.13% |
| $S_3T_4$       | 30.5  | 30.97                                         | 31.18 | 31.16 | 31.70 | 1.54%    | 2.21%   | 2.16% | 3.92% |
| $S_3T_5$       | 23.7  | 23.05                                         | 24.32 | 24.06 | 26.06 | 2.77%    | 2.59%   | 1.49% | 9.95% |