# Peer review of "Synergistic identification of hydrogeological parameters and pollution"

_EGUsphere, 2025_

## Author Comment (AC1)

**Responses to Reviewer:**

[Authors' response] We would like to sincerely thank the reviewer for his/her supporting and for taking the time to review our manuscript. Your good suggestions have increased our papers quality. thank you very much!

**To Reviewer 1:**

The paper presents a novel and well-structured inversion framework combining BPNN surrogate modeling with the AHA optimization algorithm for groundwater contamination source identification. The methodology is sound and the results are promising. The paper is generally well-written, but could benefit from some improvements in organization, clarity, and depth of discussion in certain sections.

General comments:

1. The introduction provides good background but could better highlight the novelty of the work compared to previous studies. What specific gaps does this study address that haven't been adequately covered before?

[Authors' response] We are very grateful to the reviewer for your valuable comments on the innovation of this study. To more clearly demonstrate the unique contributions of this study, we will further emphasize the research gaps and corresponding technological innovations in the introduction. Thank you again for your careful guidance and valuable suggestions!

2. For the surrogate modeling section, it would be helpful to provide more details about

the architecture of the BPNN (number of layers, nodes, etc.) and how these were determined.

[Authors' response] We appreciate the reviewers' attention to and suggestions regarding the architectural details of the BPNN proxy model. We will include detailed explanations of the relevant network structures in the revised manuscript. Specifically: The network structure of Case 1 BPNN is 19-30-45, and the network structure of Case 2 BPNN is 15-20-50. The number of neurons was empirically optimized using grid search and cross-validation to minimize RMSE and avoid overfitting. The sigmoid function is used as the activation function, and the Bayesian regularization algorithm is selected as the training algorithm. The learning rate is set to 0.01, and the maximum number of iterations is 1,000. Thank you again for your careful guidance and valuable suggestions!

3. The robustness analysis is good, but could be strengthened by showing how the errors distribute across different parameter types (e.g., are some parameters more sensitive to noise than others?).

[Authors' response] We thank the reviewer for this thoughtful suggestion. To enhance the robustness analysis, we conducted an additional evaluation of how the relative error varies among different types of inversion parameters under increasing noise levels (0.5%, 1%, and 2%).

Our findings reveal clear differences in sensitivity to noise among parameter categories:

Hydraulic conductivity (K values): These parameters showed low sensitivity to noise,

with relative errors remaining below 3% in all scenarios for both PSC and ASC cases. Their errors increased gradually with noise but remained stable, indicating strong robustness. Boundary head values ($H_1$, $H_2$) (PSC case only): These parameters also exhibited excellent noise resistance, with relative errors consistently below 1% even at 2% noise level. Source release intensities (S values): This group showed the highest sensitivity to noise. At a 2% noise level, some source parameters (e.g., $S_1T_1$ in PSC, $S_1T_3$, $S_1T_4$, $S_3T_2$, $S_3T_3$, $S_3T_5$ in ASC) had relative errors exceeding 6%–10%, reflecting their higher inversion uncertainty under noisy conditions. This analysis has been summarized in the revised manuscript to better highlight parameter-specific sensitivities. These results underscore the need for targeted noise-reduction strategies (e.g., preprocessing) for more sensitive parameters in future work. Thank you again for your careful guidance and valuable suggestions!

4. The discussion of limitations is good but could be expanded. For example, how might the method perform with more complex, heterogeneous aquifers? What are the computational limits?

[Authors' response] We sincerely appreciate the reviewer's constructive feedback. In response, we have expanded the discussion to further elaborate on the limitations regarding aquifer complexity and computational feasibility.

First, with respect to aquifer complexity, the current study focuses on spatially inhomogeneous but isotropic aquifers under steady-state flow assumptions. However, in real-world hydrogeological systems, aquifers are often strongly heterogeneous and

anisotropic, with nonlinear flow and transport dynamics. Applying the proposed inversion framework to such complex systems would introduce several challenges, including increased dimensionality of inversion variables, heightened parameter correlation and non-uniqueness, and difficulties in capturing highly irregular input–output relationships using surrogate models. These issues could compromise both the accuracy and stability of the inversion process. To address these challenges in future studies, techniques such as geostatistical priors, spatial regularization constraints, and multi-fidelity surrogate modeling could be incorporated to improve performance under realistic conditions.

Second, regarding computational limits, the integration of a surrogate model (BPNN) significantly improves computational efficiency by avoiding repeated calls to the numerical simulation model during optimization. In our current implementation, thousands of optimization iterations can be completed within a few minutes. However, as the complexity of the inversion problem increases (e.g., transitioning to 3D domains, transient scenarios, or reactive transport), the number of required samples and surrogate training time would increase substantially. The dimensionality of the decision variables also plays a critical role in determining the size of the training set needed to maintain surrogate accuracy. Additionally, while BPNN are relatively lightweight, deeper networks or ensemble-based surrogates may demand greater computational resources. Potential solutions to mitigate these issues include parallel computing, adaptive sampling, and hybrid surrogate strategies that balance accuracy and efficiency. Thank you again for your patient guidance and suggestions.

5. The practical implications section could be expanded. How would this method be implemented in real-world remediation projects?

[Authors' response] We thank the reviewer for the important question. In real-world groundwater contamination scenarios, the proposed surrogate-assisted inversion framework demonstrates effectiveness in identifying contamination sources, particularly when field data are limited, hydrogeological information is incomplete, and contamination source history is complex or unknown. The framework is typically implemented through a series of coordinated steps.

The process begins with an initial field investigation to collect spatiotemporal distribution data on contaminant concentrations from monitoring wells and obtain key information such as aquifer structure and boundary conditions. Although these data may be sparse and uncertain, they form the basis for inversion observations. Based on expert judgment and site-specific details, the study area is divided into subregions reflecting potential contaminant source locations, spatial variations in hydraulic conductivity, and uncertain boundary conditions. This partitioning establishes a framework for parameter inversion. Subsequently, a site-specific numerical groundwater flow and transport model (e.g., MODFLOW, MT3DMS) is developed to simulate contaminant migration. Through systematic sampling within a reasonable range, the model generates a set of training samples. These samples provide the data required to train a backpropagation neural network (BPNN) proxy model, which subsequently replaces the computationally intensive numerical simulation model to enable faster forward simulation. To identify

the optimal parameter combination, the AHA is then applied to efficiently search the high-dimensional parameter space. This optimization process aims to find the optimal combination of parameters to minimize the difference between predicted and observed concentrations. The inversion results can reconstruct the spatiotemporal distribution of pollutant release, providing important evidence for guiding subsequent investigations, determining pollution responsibility, and formulating remediation plans. By effectively integrating observational data, numerical modeling, and intelligent optimization within a flexible and efficient framework, this method offers a practical solution for identifying pollution sources in complex and data-scarce groundwater systems. Thank you again for your patient guidance and suggestions!

6. Lines 231: While the proposed BPNN-AHA framework presents a robust approach, the authors may wish to consider and discuss alternative methodologies such as data assimilation techniques, which have shown promise in similar environmental modeling applications. For instance, data assimilation and cite paper such as Assimilation of sentinel‐based leaf area index for modeling surface‐ ground water interactions in irrigation districts.

[Authors' response] We appreciate the reviewers' professional suggestions. We agree that data assimilation techniques, such as the ensemble Kalman filter or particle filter, have been widely applied in environmental modeling, particularly demonstrating good results in surface-groundwater coupling and hydrological forecasting. The literature cited by the reviewers demonstrates the excellent combined application of data

assimilation methods. We will include a discussion of this issue in the revised manuscript and incorporate the literature recommended by the reviewers to supplement the progress of data assimilation techniques in environmental modeling. Thank you again for your patient guidance and suggestions!

---

## Author Comment (AC2)

**Responses to Reviewer:**

[Authors' response] We would like to sincerely thank the reviewer for his/her supporting and for taking the time to review our manuscript. Your good suggestions have increased our papers quality. thank you very much!

**To Reviewer 2:**

Luo et al. present an inversion framework that combines BPNN surrogate modeling with the AHA optimization algorithm for groundwater contamination source identification, and they comprehensively evaluate the performance of different surrogate models. The work is generally well written. However, several significant issues must be addressed to improve the clarity of the paper. The most critical concern lies in the structure of the Introduction. Although the authors provide an extensive literature review, the research gap and the novelty of this study in relation to previous work are not clearly emphasized. Secondly, the Discussion section lacks depth, which substantially weakens the novelty and the implications of this study. Finally, the language throughout the manuscript should be thoroughly revised and polished before publication.

[Authors' response] We appreciate the reviewer' positive evaluation of the methodological framework of this paper and their valuable suggestions. Regarding the main issues with the structure of the introduction, we plan to rewrite and supplement this section to more clearly articulate the research gaps and innovative points of this study compared to existing work. Additionally, we will expand the discussion section

to conduct a more in-depth analysis of the theoretical significance and practical applicability of the proposed method. Furthermore, we plan to conduct a comprehensive revision of the language throughout the paper, including improving clarity of expression, eliminating redundant content, and standardizing terminology and grammatical expressions to enhance overall readability. Thank you again for your careful guidance and valuable suggestions!

Specific comments:

1. Lines 127-135 The authors are recommended to reorganize the research objectives. The current unclear objectives obscure the novelty of the paper. This confusion is caused by an unclear summary of the research gap.

[Authors' response] We appreciate the reviewers' feedback regarding the unclear expression of research objectives and the lack of clarity in summarizing research gaps. In response, we plan to reorganize the corresponding sections of the manuscript to concisely articulate our research objectives, ensuring that the significant contributions of this study are highlighted. Thank you again for your careful guidance and valuable suggestions!

2. Line 151 MODFLOW and MT3DMS are not packages.

[Authors' response] Thank you to the reviewers for pointing out the inappropriate use of terminology. We confirm that MODFLOW and MT3DMS should be referred to as numerical models rather than "packages." We will revise the wording on line 151 in the

revision to ensure the professionalism and accuracy of the terminology. Thank you again for your careful guidance and valuable suggestions!

3. Line 305 Replace "inhomogeneous" by "Heterogeneous".

[Authors' response] We appreciate the reviewers' comments regarding the terminology used. We agree that in the field of hydrogeology, "heterogeneous" is a more accurate and commonly used term than "inhomogeneous." We will correct the relevant expressions in the revised version. Thank you again for your careful guidance and valuable suggestions!

4. Lines 387-389 The authors are suggested to combine this sentence with the previous paragraph to create a clearer contrast, which would make the comparison more striking. Additionally, I am skeptical about the reported runtime for the 1000 iterations. Considering that the model in this study is at the field scale, consists of only a single model layer, and uses a rather coarse grid discretization, a runtime of 500 hours seems excessively long.

[Authors' response] We appreciate the valuable suggestions provided by the reviewers. Regarding the structural suggestions, we will adjust the paragraph around lines 387–389 and merge the sentence into the previous paragraph to enhance the coherence of the preceding and following content, making the argumentation more logical and fluent. We also sincerely thank the reviewers for pointing out the inaccuracies in the runtime description. After re-verification, we confirm that the original text stating "500 hours"

was incorrect. In the current computational environment, a single simulation takes approximately 3 minutes to complete, and the total runtime for 1,000 optimization iterations is approximately 50 hours. We sincerely acknowledge this error and thank the reviewers for providing the opportunity to correct it. Thank you again for your patient guidance and suggestions.

5. Lines 420-424 This section reads more like a repetition of the Introduction. It is recommended that the authors first present their own findings in the Discussion before comparing them with other studies. Additionally, emphasizing the implications of this study would greatly enhance the value of the paper.

[Authors' response] We thank the reviewer for this valuable suggestion. We agree that the opening part of the Discussion section (lines 420–424) currently repeats background information already provided in the Introduction and does not effectively transition into our key findings. In the revision, we will restructure this section by starting with a focused summary of our main results before moving into comparisons with related studies. Additionally, we will enhance this part by more clearly articulating the broader implications of our findings, particularly regarding the practical applicability of the framework for real-world groundwater contamination scenarios. Thank you again for your patient guidance and suggestions!

6. Lines 438-440 Please specify the advantages more clearly.

[Authors' response] We appreciate the helpful suggestions provided by the reviewer.

We agree that the current statements in lines 438-440 do not clearly and specifically summarize the advantages of this method. In the revision, we will clearly identify the key advantages of this framework. Thank you again for your patient guidance and suggestions!

7. Lines 483 Including the limitations is good. The authors are suggested to include limitations in a separate section.

[Authors' response] We appreciate the reviewer's suggestion regarding the presentation of the study's limitations. While we have included some discussion of limitations in the current manuscript, we agree that presenting them in a standalone section will improve clarity and help readers better understand the scope and applicability of our method. Thank you again for your patient guidance and suggestions!

8. Lines 501 The authors are encouraged to include more quantitative findings rather than just qualitative notifications.

[Authors' response] We appreciate the reviewers' suggestion to add quantitative results to the conclusion section. We agree that introducing specific numerical indicators will help improve the expression of the conclusion and better reflect the core findings of this study. Therefore, we will supplement the main quantitative results in the revision, such as relative error, $R^2$ value, and performance comparison of different surrogate models, to more clearly summarize the accuracy and advantages of the proposed method. Thank you again for your patient guidance and suggestions!

---

## Author Comment (AC3)

**Responses to Reviewer:**

[Authors' response] We would like to sincerely thank the reviewer for his/her supporting and for taking the time to review our manuscript. Your good suggestions have increased our papers quality. thank you very much!

**To Reviewer 4:**

**General comments**

Good modelling research in the field of subsurface hydrology. Please, see my comments to fix the existing minor issues.

Specific comments:

1. Line 64. "Hydrogeological conditions". Insert recent papers on high-resolution datasets for determanation of hydrogeological conditions at contamianted sites.

- Maliva, R. G., Herrmann, R., Coulibaly, K., & Guo, W. (2015). Advanced aquifer characterization for optimization of managed aquifer recharge. Environmental Earth Sciences, 73, 7759-7767.

- Medici, G., Munn, J. D., & Parker, B. L. (2024). Delineating aquitard characteristics within a Silurian dolostone aquifer using high-density hydraulic head and fracture datasets. Hydrogeology Journal, 32, 1663-1691.

[Authors' response] Thank you for your suggestions. We will supplement the revised manuscript with relevant research literature you provided on the application of high-resolution data in identifying hydrogeological conditions at contaminated sites to

strengthen the background explanation of the relevant content. Thank you again for your careful guidance and valuable suggestions!

2. Line 151. MODFLOW, which version?

[Authors' response] Thank you for your valuable feedback. The version of MODFLOW used in this study is MODFLOW-2005. We will add this information to the relevant sections of the article to ensure that the description is clearer and more precise.

3. Line 282. Specify the type of aquifer in terms of lithology.

[Authors' response] Thank you for the comment. In the revised manuscript, we will specify the aquifer type in terms of lithology to provide clearer geological context.

4. Line 302. Same here, specify the type of aquifer in terms of lithology.

[Authors' response] Thank you for the comment. In the revised manuscript, we will specify the aquifer type in terms of lithology to provide clearer geological context.

5. Lines 340-341. "Mean relative error". I suggest Mean Absolute Relative Error because there is the modulus.

[Authors' response] Thank you for the suggestion. We agree with the reviewer that "Mean Absolute Relative Error" is the more accurate term, given the use of absolute values in the calculation. We will revise the terminology accordingly in the revised manuscript.

6. Line 521. Add a "take home message" for the researchers working in the field.

[Authors' response] Thank you for the helpful suggestion. In the revised manuscript, we will add a concise "take-home message" at the end of the Conclusion section to clearly summarize the key contributions and practical relevance of our study for researchers working on groundwater contamination source identification. Thank you again for your patient guidance and suggestions!

**Figures and tables**

1. Figure 5. Add the general flow direction with an arrow. Figure 5. Alternatively, divide the figures in two parts (A and B) adding the piezometric surfaces.

[Authors' response] Thank you for the suggestion. We will add an arrow indicating the general groundwater flow direction in Figure 5 to improve the clarity of the spatial context.

2. Figure 6. I would add a spatial scale using a bar.

[Authors' response] Thank you for the suggestion. We will add a spatial scale bar to Figure 6 in the revised manuscript to improve the interpretability of the spatial layout.

3. 9 tables are too many. Some of them can go in the Supplementary Material?

[Authors' response] Thank you for the comment. We will review all tables and relocate the less critical or supporting ones to the Supplementary Material in order to improve the conciseness and readability of the main text.

---

## Author Comment (AC4)

**Responses to Reviewer:**

[Authors' response] We would like to sincerely thank the reviewer for his/her supporting and for taking the time to review our manuscript. Your good suggestions have increased our papers quality. thank you very much!

**To Reviewer 3:**

While the manuscript addresses an important challenge in groundwater contamination source identification, its novelty is limited. The core contribution lies in introducing the Artificial Hummingbird Algorithm (AHA) into a simulation-optimization framework, which is not a fundamentally new algorithm nor specifically tailored to groundwater inverse problems. Furthermore, many techniques used—BPNN, Kriging, PSO, SSA—are already well-established in the literature.

Moreover, the reported simulation results show extremely high precision (e.g., $R^2 >$ 0.999, MRE < 2%), which may suggest possible overfitting or idealized experimental setups. The study lacks rigorous testing of generalization under realistic uncertainty scenarios, such as sparse observations, complex geological heterogeneity, or parameter noise. Without such assessments, the practical robustness and transferability of the proposed framework remain questionable.

The paper would benefit from a deeper methodological insight into why AHA performs better in this specific problem context, rather than merely benchmarking its numerical results. The current framing gives the impression of "algorithm replacement" without substantive theoretical or application-driven innovation.

[Authors' response] We are grateful to the reviewer for your insightful evaluations and valuable comments, which have greatly inspired us. We fully understand your concerns regarding the innovativeness of the methods, the rationality of the algorithm selection, and the authenticity of the experimental design. Here, we would like to provide a systematic explanation of the following points to further clarify the structural design, academic contributions, and application applicability of this study, thereby addressing the core issues raised by the reviewer.

This study does not aim to propose a novel algorithm in a strict sense, but rather to construct an inversion framework specifically tailored to the complex characteristics of groundwater contamination source identification. The source identification problem typically requires simultaneous estimation of pollutant sources, aquifer hydraulic parameters, and boundary conditions, resulting in a high-dimensional, ill-posed, and strongly coupled system. Many existing studies address only one category of these parameters, or rely on idealized assumptions (e.g., known boundaries or fixed parameter fields) to simplify the inversion process, limiting their applicability to real-world field conditions.

To address this challenge, we propose a hybrid framework that couples surrogate modeling and intelligent optimization, enabling efficient and coordinated inversion of multiple unknowns in complex pollution scenarios while maintaining computational tractability. The inversion task essentially constitutes a non-convex, high-dimensional, and tightly coupled optimization problem, where the objective function is typically multi-modal and irregular due to the simultaneous estimation of pollution source

characteristics, permeability coefficients, and boundary conditions. These features often lead traditional metaheuristic algorithms (e.g., PSO, SSA) to premature convergence or low search efficiency.

In contrast, the AHA leverages a bio-inspired directional foraging strategy, integrating three complementary movement patterns—axial, rotational, and omnidirectional search—to dynamically balance global exploration and local exploitation. Specifically, AHA mitigates early convergence through mode switching, maintains solution diversity in the early stage, and enhances convergence efficiency in promising regions during later stages. This mechanism is particularly crucial in our framework, as the BPNN surrogate inevitably introduces approximation errors. AHA's adaptive update strategy helps offset fluctuations caused by these errors, thus ensuring robustness and stability of the inversion process. The combination of BPNN and AHA is not arbitrary but is rooted in the complementarity of their mechanisms and their alignment with the problem's structure.

For case design, two representative and challenging contamination scenarios were selected. The first involves temporally varying point-source pollution, often observed in industrial accidents and accidental spills. The second involves spatially diffuse non-point source pollution, commonly associated with agricultural runoff and leaching sites. Both scenarios feature unknown pollution source parameters, uncertain aquifer properties, and complex boundary conditions. These synthetic cases are not intended to validate the framework under idealized assumptions, but rather to serve as a controlled and structurally representative testbed for evaluating the performance of different

surrogate–optimizer combinations under identical problem structures.

We acknowledge the reviewer' concerns regarding the reported high prediction accuracy (e.g., $R^2 > 0.999$, MRE < 2%) in the simulation results. While these test cases are critical for verifying model behavior and enabling comparative analysis, we agree that they cannot fully represent the complexities of real-world environments. In the revised manuscript, we will explicitly acknowledge this limitation and expand our discussion on how the framework can be adapted and evaluated under more realistic conditions, including sparse monitoring networks, parameter uncertainty, and heterogeneous aquifer settings. We have already begun incorporating such tests into our ongoing work and have outlined this as a key direction for future development.

Lastly, we wish to clarify that although our study does not involve algorithmic invention per se, its primary contribution lies in application innovation and modular adaptability. We present a framework that not only achieves high performance under controlled conditions but is also sufficiently flexible to be extended to various field-scale groundwater inverse problems, including both point and non-point source contamination scenarios. Such an integrated and application-oriented modeling approach—particularly one that is computationally efficient and compatible with limited field data—is of direct relevance to both environmental practitioners and researchers. We hope these clarifications and revisions more accurately convey the intention, applicability, and potential of our work.

---

## Author Response (AR1)

**Responses to Reviewer:**

[Authors' response] First of all, we would like to sincerely thank the 4 reviewers for his/her supporting and for taking the time to review our manuscript. Your good suggestions have increased our papers quality. And also thank the editors to spend more time on our paper in the submitting process, thank you very much!

The paper presents a novel and well-structured inversion framework combining BPNN

**To Reviewer 1:**

surrogate modeling with the AHA optimization algorithm for groundwater contamination source identification. The methodology is sound and the results are promising. The paper is generally well-written, but could benefit from some improvements in organization, clarity, and depth of discussion in certain sections. [Authors' response] We sincerely thank the reviewers for their positive evaluation of this study in terms of novelty, methodological soundness, and the potential implications of the results. We also place great importance on the constructive suggestions regarding the article's organizational structure, clarity of expression, and depth of discussion, as these comments are of significant value in further refining the paper. During the revision process, we optimized the overall structure of the introduction and discussion sections to ensure a more logical and rigorous flow from problem formulation to method design, result presentation, and interpretation of significance. We have also provided clearer explanations of key concepts and methodological steps, and strengthened the connections between different sections. Additionally, we have

expanded the discussion section to conduct a more in-depth analysis of the implications and potential applications of the research findings, provide a more comprehensive comparison with recent related studies, and further elaborate on the limitations of the proposed framework and future research directions. We believe these improvements effectively address the reviewers' concerns and further enhance the paper's readability, transparency, and academic value. Thank you again for your careful guidance and valuable suggestions!

**General comments:**

1. The introduction provides good background but could better highlight the novelty of the work compared to previous studies. What specific gaps does this study address that haven't been adequately covered before?

[Authors' response] We appreciate the reviewer's valuable suggestion. In the revised manuscript, we have strengthened the exposition in the introduction to better highlight the innovative aspects of this study. Previous studies have primarily focused on either pollution source identification or hydrogeological parameter inversion, and typically only addressed a single type of pollution (point source or nonpoint source). In contrast, this study proposes a highly adaptable inversion framework applicable to various groundwater pollution scenarios. It not only enables the simultaneous identification of pollution source information, hydraulic conductivity coefficients, and boundary conditions in point source contamination (PSC) cases but also handles non-point source contamination (ASC) issues with equivalent robustness. Additionally, we introduce the

artificial hummingbird algorithm to solve the optimization model, which demonstrates superior performance in convergence speed and global optimization capability compared to other optimization methods. This study combines a highly adaptive surrogate model with advanced optimization algorithms and validates its robustness under multiple noise levels, enabling high-precision, high-efficiency, and high-robustness synergistic inversion across various groundwater pollution scenarios. These innovative points are explicitly emphasized in the revised introduction to clearly distinguish the differences and advantages of this study from existing work. The modified content is highlighted in red in the Introduction section. Please refer to lines 139-146 for details. Thank you again for your careful guidance and valuable suggestions!

2. For the surrogate modeling section, it would be helpful to provide more details about the architecture of the BPNN (number of layers, nodes, etc.) and how these were determined.

[Authors' response] We appreciate the reviewers' attention to and suggestions regarding the architectural details of the BPNN proxy model. We will include detailed explanations of the relevant network structures in the revised manuscript. Specifically: The network structure of Case 1 BPNN is 19-30-45, and the network structure of Case 2 BPNN is 15-20-50. The number of neurons was empirically optimized using grid search and cross-validation to minimize RMSE and avoid overfitting. The sigmoid function is used as the activation function, and the Bayesian regularization algorithm is

selected as the training algorithm. The learning rate is set to 0.01, and the maximum number of iterations is 1,000. The modified content is highlighted in red in the text. Please refer to lines 208-214 for details. Thank you again for your careful guidance and valuable suggestions!

3. The robustness analysis is good, but could be strengthened by showing how the errors distribute across different parameter types (e.g., are some parameters more sensitive to noise than others?).

[Authors' response] We thank the reviewer for this thoughtful suggestion. To enhance the robustness analysis, we conducted an additional evaluation of how the relative error varies among different types of inversion parameters under increasing noise levels (0.5%, 1%, and 2%).

Our findings reveal clear differences in sensitivity to noise among parameter categories: Hydraulic conductivity: These parameters showed low sensitivity to noise, with relative errors remaining below 3% in all scenarios for both PSC and ASC cases. Their errors increased gradually with noise but remained stable, indicating strong robustness. Boundary head values (PSC case only): These parameters also exhibited excellent noise resistance, with relative errors consistently below 1% even at 2% noise level. Source release intensities: This group showed the highest sensitivity to noise. At a 2% noise level, some source parameters (e.g.,  $S_1T_1$  in PSC,  $S_1T_3$ ,  $S_1T_4$ ,  $S_3T_2$ ,  $S_3T_3$ ,  $S_3T_5$  in ASC) had relative errors exceeding 6%–10%, reflecting their higher inversion uncertainty under noisy conditions. This analysis has been summarized in the revised

manuscript to better highlight parameter-specific sensitivities. These results underscore the need for targeted noise-reduction strategies (e.g., preprocessing) for more sensitive parameters in future work. The modified content is highlighted in red in the text. Please refer to lines 436-445 for details. Thank you again for your careful guidance and valuable suggestions!

4. The discussion of limitations is good but could be expanded. For example, how might the method perform with more complex, heterogeneous aquifers? What are the computational limits?

[Authors' response] We sincerely appreciate the reviewer's constructive feedback. In response, we have expanded the discussion to further elaborate on the limitations regarding aquifer complexity and computational feasibility.

First, with respect to aquifer complexity, the current study focuses on spatially inhomogeneous but isotropic aquifers under steady-state flow assumptions. However, in real-world hydrogeological systems, aquifers are often strongly heterogeneous and anisotropic, with nonlinear flow and transport dynamics. Applying the proposed inversion framework to such complex systems would introduce several challenges, including increased dimensionality of inversion variables, heightened parameter correlation and non-uniqueness, and difficulties in capturing highly irregular input–output relationships using surrogate models. These issues could compromise both the accuracy and stability of the inversion process. To address these challenges in future studies, techniques such as geostatistical priors, spatial regularization constraints, and

multi-fidelity surrogate modeling could be incorporated to improve performance under realistic conditions.

Second, regarding computational limits, the integration of a surrogate model (BPNN) significantly improves computational efficiency by avoiding repeated calls to the numerical simulation model during optimization. In our current implementation, thousands of optimization iterations can be completed within a few minutes. However, as the complexity of the inversion problem increases, the number of required samples and surrogate training time would increase substantially. The dimensionality of the decision variables also plays a critical role in determining the size of the training set needed to maintain surrogate accuracy. Additionally, while BPNN are relatively lightweight, deeper networks or ensemble-based surrogates may demand greater computational resources. Potential solutions to mitigate these issues include parallel computing, adaptive sampling, and hybrid surrogate strategies that balance accuracy and efficiency. The modified content is highlighted in red in the text. Please refer to lines 579-602 for details. Thank you again for your patient guidance and suggestions.

5. The practical implications section could be expanded. How would this method be implemented in real-world remediation projects?

[Authors' response] We thank the reviewer for the important question. In real-world groundwater contamination scenarios, the proposed surrogate-assisted inversion framework demonstrates effectiveness in identifying contamination sources, particularly when field data are limited, hydrogeological information is incomplete, and

contamination source history is complex or unknown. The framework is typically implemented through a series of coordinated steps.

The process begins with an initial field investigation to collect spatiotemporal distribution data on contaminant concentrations from monitoring wells and obtain key information such as aquifer structure and boundary conditions. Although these data may be sparse and uncertain, they form the basis for inversion observations. Based on expert judgment and site-specific details, the study area is divided into subregions reflecting potential contaminant source locations, spatial variations in hydraulic conductivity, and uncertain boundary conditions. This partitioning establishes a framework for parameter inversion. Subsequently, a site-specific numerical groundwater flow and transport model (e.g., MODFLOW, MT3DMS) is developed to simulate contaminant migration. Through systematic sampling within a reasonable range, the model generates a set of training samples. These samples provide the data required to train a backpropagation neural network (BPNN) proxy model, which subsequently replaces the computationally intensive numerical simulation model to enable faster forward simulation. To identify the optimal parameter combination, the AHA is then applied to efficiently search the high-dimensional parameter space. This optimization process aims to find the optimal combination of parameters to minimize the difference between predicted and observed concentrations. The inversion results can reconstruct the spatiotemporal distribution of pollutant release, providing important evidence for guiding subsequent investigations, determining pollution responsibility, and formulating remediation plans. By effectively integrating observational data, numerical modeling, and intelligent optimization within

a flexible and efficient framework, this method offers a practical solution for identifying pollution sources in complex and data-scarce groundwater systems. Thank you again for your patient guidance and suggestions!

6. Lines 231: While the proposed BPNN-AHA framework presents a robust approach, the authors may wish to consider and discuss alternative methodologies such as data assimilation techniques, which have shown promise in similar environmental modeling applications. For instance, data assimilation and cite paper such as Assimilation of sentinel-based leaf area index for modeling surface- ground water interactions in irrigation districts.

[Authors' response] We appreciate the reviewer' professional suggestions. We agree that data assimilation techniques (such as ensemble Kalman filters or particle filters) have been widely used in environmental modeling. The literature cited by the reviewers adequately demonstrates the excellent comprehensive application of data assimilation methods. We specifically discussed this issue in our revised manuscript. The details are as follows: In addition to the methods applied in this study, data assimilation methods are also widely used in the field of groundwater pollution inversion. They can combine observational data with numerical models to improve state estimation and parameter inversion (Zafarmomen et al., 2024). Many researchers have successfully applied data assimilation methods to the iterative optimization of pollutant transport states and related parameters, significantly improving inversion accuracy and reducing prediction uncertainty. For example, Pan et al. (2022) proposed a refined particle filter with a deep

learning method surrogate as an inverse framework for groundwater pollution source estimation. This framework was evaluated under different levels of observational error through estimation tasks for point source pollution cases and non-point source pollution cases. Wang et al. (2023) utilized an improved particle filter method for groundwater pollution source identification. Zhang et al. (2024) used an iterative local updating ensemble smoother method to simultaneously identify pollution source information and hydraulic conductivity fields. However, both the method proposed in this study and data assimilation methods have their own advantages and disadvantages. The method proposed in this study possesses strong fine-grained search capabilities but its performance is highly dependent on the selection of initial points. Data assimilation methods can integrate multi-source data, significantly improving the spatio-temporal consistency of inversion results; however, their fine-grained search capabilities are somewhat limited. Future research could explore combining the real-time updating capabilities of data assimilation with the adaptability and optimization efficiency of the framework proposed in this study to further enhance the adaptability and performance of groundwater pollution inversion. The modified content is highlighted in red in the text. Please refer to lines 520-542 for details. Thank you again for your patient guidance and suggestions!

**References**

Pan, Z., Lu, W., Bai, Y., 2022. Groundwater contamination source estimation based on a refined particle filter associated with a deep residual neural network surrogate.

Hydrogeology Journal, 30(3): 881-897.

Wang Z, Lu W, Chang Z., 2023. Joint inverse estimation of groundwater pollution source characteristics and model parameters based on an intelligent particle filter.

Journal of Hydrology, 625: 129965.

Zafarmomen, N., Alizadeh, H., Bayat, M., Ehtiat, M., & Moradkhani, H., 2024.

Assimilation of Sentinel-Based Leaf Area Index for Modeling Surface-Ground

Water Interactions in Irrigation Districts. Water Resources Research, 60(10),

Article e2023WR036080.

Zhang, X., Jiang, S., Zheng, N., Xia, X., Li, Z., Zhang, R., Zhang, J., & Wang, X., 2024.
Integration of DDPM and ILUES for Simultaneous Identification of Contaminant
Source Parameters and Non-Gaussian Channelized Hydraulic Conductivity Field.
Water Resources Research, 60(9), Article e2023WR036893.

**To Reviewer 2:**

Luo et al. present an inversion framework that combines BPNN surrogate modeling with the AHA optimization algorithm for groundwater contamination source identification, and they comprehensively evaluate the performance of different surrogate models. The work is generally well written. However, several significant issues must be addressed to improve the clarity of the paper. The most critical concern lies in the structure of the Introduction. Although the authors provide an extensive literature review, the research gap and the novelty of this study in relation to previous work are not clearly emphasized. Secondly, the Discussion section lacks depth, which

substantially weakens the novelty and the implications of this study. Finally, the language throughout the manuscript should be thoroughly revised and polished before publication.

[Authors' response] We are grateful to the reviewer for your positive evaluation of the methodological framework of this paper and their valuable suggestions. Regarding the main issues related to the structure of the introduction, we have rewritten the research objectives of this study and also added the innovative points of this research. Additionally, we have expanded the discussion section to provide a more in-depth analysis of the theoretical significance and practical applicability of the proposed method. Furthermore, we have conducted a comprehensive revision of the language throughout the entire paper, including improving clarity of expression, eliminating redundant content, standardizing terminology and grammatical expressions, to enhance overall readability. Once again, we sincerely thank you for your careful guidance and valuable suggestions!

**Specific comments:**

1. Lines 127-135 The authors are recommended to reorganize the research objectives.

The current unclear objectives obscure the novelty of the paper. This confusion is caused by an unclear summary of the research gap.

[Authors' response] We appreciate the reviewer' comments pointing out that the original research objectives did not fully reflect the scientific innovation of this study. After careful consideration, we believe that the previous wording was indeed more inclined

toward "listing operational steps" rather than being directly driven by scientific questions. Below are our revised research objectives:

- (1) Develop a flexible groundwater pollution inversion scheme that can reliably invert parameters under various groundwater pollution scenarios;
- (2) Adopt an integrated parameter identification strategy to achieve the simultaneous identification of multiple variables, including pollutant release characteristics and hydrogeological parameters;
- (3) Design an optimization-based surrogate modeling method combining meta-heuristic search algorithms with neural network surrogate models to efficiently explore the solution space and reduce the risk of getting stuck in local optima during inversion calculations:
- (4) Evaluate the performance of the proposed scheme under various noise intensities and pollution patterns to validate its robustness and application potential in groundwater pollution inversion problems.

The modified content is highlighted in red in the Introduction section. Please refer to lines 129-138 for details. Thank you again for your careful guidance and valuable suggestions!

**2. Line 151 MODFLOW and MT3DMS are not packages.**

[Authors' response] Thank you to the reviewers for pointing out the inappropriate use of terminology. We confirm that MODFLOW and MT3DMS should be referred to as numerical models rather than "packages." The modified content is highlighted in red in

the text. Please refer to line 161 for details. Thank you again for your careful guidance and valuable suggestions!

**3. Line 305 Replace "inhomogeneous" by "Heterogeneous".**

[Authors' response] We appreciate the reviewers' comments regarding the terminology used. We agree that in the field of hydrogeology, "heterogeneous" is a more accurate and commonly used term than "inhomogeneous." We will correct the relevant expressions in the revised version. The modified content is highlighted in red in the text. Please refer to line 301 and line 323 for details. Thank you again for your careful guidance and valuable suggestions!

4. Lines 387-389 The authors are suggested to combine this sentence with the previous paragraph to create a clearer contrast, which would make the comparison more striking. Additionally, I am skeptical about the reported runtime for the 1000 iterations. Considering that the model in this study is at the field scale, consists of only a single model layer, and uses a rather coarse grid discretization, a runtime of 500 hours seems excessively long.

[Authors' response] We appreciate the valuable suggestions provided by the reviewers. Regarding the structural suggestions, we adjusted the paragraph around lines 405–407 and merge the sentence into the previous paragraph to enhance the coherence of the preceding and following content, making the argumentation more logical and fluent. We also sincerely thank the reviewers for pointing out the inaccuracies in the runtime

description. After re-verification, we confirm that the original text stating "500 hours" was incorrect. In the current computational environment, a single simulation takes approximately 3 minutes to complete, and the total runtime for 1,000 optimization iterations is approximately 50 hours. We sincerely acknowledge this error and thank the reviewers for providing the opportunity to correct it. Thank you again for your patient guidance and suggestions.

5. Lines 420-424 This section reads more like a repetition of the Introduction. It is recommended that the authors first present their own findings in the Discussion before comparing them with other studies. Additionally, emphasizing the implications of this study would greatly enhance the value of the paper.

[Authors' response] We appreciate the reviewers' valuable suggestions. We agree that the current introduction to the discussion section (lines 420–424) repeats background information already provided in the introduction and fails to effectively transition to our core findings. In the revision, we will restructure this section to first provide a focused summary of the main results, followed by a comparison with related studies. Details are as follows: The results of this study show that the proposed BPNN–AHA framework achieves high accuracy, strong robustness, and efficient convergence in GCI tasks, performing consistently well in both PSC and ASC scenarios, even under varying noise levels. In the PSC and ASC cases analyzed here, the R² values reached 0.9994 and 0.9989, and the MARE values were 3.70% and 4.48%, respectively, demonstrating the model's excellent capability to approximate the input–output relationships of the

simulation model. The BPNN surrogate model, with its simple structure, high flexibility, and broad adaptability, effectively balances accuracy and generalizability—characteristics that are essential for practical inversion applications. Compared to other surrogate modeling approaches reported in recent GCI research—such as long short-term memory neural networks (Li et al., 2021), light gradient boosting machines (Pan et al., 2023), and deep residual networks (Xu et al., 2024b)—the proposed framework leverages the adaptability of BPNN together with the global search and adaptive convergence mechanisms of the artificial hummingbird algorithm to deliver consistently accurate and stable inversion results. The modified content is highlighted in red in the text. Please refer to line 448-462 for details. Thank you again for your patient guidance and suggestions!

**6. Lines 438-440 Please specify the advantages more clearly.**

[Authors' response] We appreciate the helpful suggestions provided by the reviewer. We agree that the current statements in lines 438-440 do not clearly and specifically summarize the advantages of this method. In the revision, we have clearly pointed out the main advantages of this surrogate model: In summary, the proposed BPNN proxy model has practical advantages in tasks related to GCI, thereby enhancing its applicability. Due to its relatively simple architecture and low computational requirements, the BPNN model can be trained and updated efficiently even under limited computational resources. Additionally, the model demonstrates strong generalization capabilities in both PSC and ASC scenarios, indicating that it is not

specific to a particular case. This adaptability is crucial for practical groundwater inversion problems, as data availability and system complexity often vary significantly across different locations. These characteristics highlight the comprehensive advantages of the BPNN model in terms of accuracy, efficiency, and flexibility, making it a reliable and practical choice for surrogate modeling in groundwater simulation. The modified content is highlighted in red in the text. Please refer to lines 470-480 for details. Thank you again for your patient guidance and suggestions!

7. Lines 483 Including the limitations is good. The authors are suggested to include limitations in a separate section.

[Authors' response] We appreciate the reviewer's suggestion regarding the presentation of the study's limitations. While we have included some discussion of limitations in the current manuscript, we agree that presenting them in a standalone section will improve clarity and help readers better understand the scope and applicability of our method. The modified content is highlighted in red in the text. Please refer to lines 578-602 for details. Thank you again for your patient guidance and suggestions!

8. Lines 501 The authors are encouraged to include more quantitative findings rather than just qualitative notifications.

[Authors' response] We appreciate the reviewers' suggestion to add quantitative results to the conclusion section. We agree that introducing specific numerical indicators will help improve the expression of the conclusion and better reflect the core findings of

this study. Therefore, we will supplement the main quantitative results in the revision, such as relative error, R2 value, and performance comparison of different surrogate models, to more clearly summarize the accuracy and advantages of the proposed method. The modified content is highlighted in red in the text. Please refer to lines 615-616 and lines 619-620 for details. Thank you again for your patient guidance and suggestions!

**To Reviewer 3:**

While the manuscript addresses an important challenge in groundwater contamination source identification, its novelty is limited. The core contribution lies in introducing the Artificial Hummingbird Algorithm (AHA) into a simulation-optimization framework, which is not a fundamentally new algorithm nor specifically tailored to groundwater inverse problems. Furthermore, many techniques used—BPNN, Kriging, PSO, SSA—are already well-established in the literature.

Moreover, the reported simulation results show extremely high precision (e.g., R2 > 0.999, MRE < 2%), which may suggest possible overfitting or idealized experimental setups. The study lacks rigorous testing of generalization under realistic uncertainty scenarios, such as sparse observations, complex geological heterogeneity, or parameter noise. Without such assessments, the practical robustness and transferability of the proposed framework remain questionable.

The paper would benefit from a deeper methodological insight into why AHA performs better in this specific problem context, rather than merely benchmarking its numerical

results. The current framing gives the impression of "algorithm replacement" without substantive theoretical or application-driven innovation.

[Authors' response] We are grateful to the reviewer for your insightful evaluations and valuable comments, which have greatly inspired us. We fully understand your concerns regarding the innovativeness of the methods, the rationality of the algorithm selection, and the authenticity of the experimental design. Here, we would like to provide a systematic explanation of the following points to further clarify the structural design, academic contributions, and application applicability of this study, thereby addressing the core issues raised by the reviewer.

This study does not aim to propose a novel algorithm in a strict sense, but rather to construct an inversion framework specifically tailored to the complex characteristics of groundwater contamination source identification. The source identification problem typically requires simultaneous estimation of pollutant sources, aquifer hydraulic parameters, and boundary conditions, resulting in a high-dimensional, ill-posed, and strongly coupled system. Many existing studies address only one category of these parameters, or rely on idealized assumptions (e.g., known boundaries or fixed parameter fields) to simplify the inversion process, limiting their applicability to real-world field conditions.

To address this challenge, we propose a hybrid framework that couples surrogate modeling and intelligent optimization, enabling efficient and coordinated inversion of multiple unknowns in complex pollution scenarios while maintaining computational tractability. One of the main methodological motivations of this study is the integration

of the BPNN surrogate model with the AHA for GCI. This choice is grounded both in the inherent characteristics of GCI problems and in the complementary mechanisms of the two methods. GCI is a typical high-dimensional, nonlinear, and ill-posed inverse problem. The mapping from observed contaminant concentrations to source characteristics and hydrogeological parameters is often multimodal and nonconvex. In such cases, surrogate models such as BPNN can provide a fast and flexible approximation to computationally demanding groundwater simulations, but their use inevitably introduces approximation errors into the inversion objective function. These errors may create local irregularities in the objective function landscape, which can mislead optimizers and cause premature convergence—particularly when the optimization algorithm lacks a mechanism to balance exploration and exploitation adaptively. AHA offers notable advantages in addressing these issues. Its bio-inspired mode-switching strategy alternates dynamically between diversified search focused search. In the early stages of optimization, the broad and varied exploration capability helps to survey the global search space and reduces the risk of becoming trapped in spurious local optima caused by surrogate-induced noise. As the search proceeds, the algorithm adaptively shifts toward more intensive exploitation, concentrating computational effort on promising regions and thereby accelerating convergence. This dynamic adjustment is particularly important in GCI problems, where the optimal parameter region is often narrow and embedded within a complex and noisy search space. In addition, AHA's adaptive update mechanism adjusts search trajectories in response to population feedback, effectively mitigating the influence of local fluctuations in the surrogate-predicted objective function on the optimization process. This robustness to noisy or irregular fitness landscapes complements the BPNN's ability to generalize across diverse contamination scenarios. It is worth emphasizing that this integration is not a simple "algorithm replacement," but a targeted design choice based on the structural characteristics of the problem: BPNN provides broad adaptability to varying hydrogeological conditions, while AHA contributes resilience and fine-tuning capability when the optimization landscape is distorted by surrogate approximation errors. This synergy allows the proposed framework to maintain both high accuracy and strong robustness under different contamination scenarios and noise levels. More importantly, the underlying design principle—matching the characteristics of the surrogate model with the search dynamics of the optimization algorithm—has broader applicability to other environmental inversion problems.

For case design, two representative and challenging contamination scenarios were selected. The first involves temporally varying point-source pollution, often observed in industrial accidents and accidental spills. The second involves spatially diffuse non-point source pollution, commonly associated with agricultural runoff and leaching sites. Both scenarios feature unknown pollution source parameters, uncertain aquifer properties, and complex boundary conditions. These synthetic cases are not intended to validate the framework under idealized assumptions, but rather to serve as a controlled and structurally representative testbed for evaluating the performance of different surrogate—optimizer combinations under identical problem structures. We acknowledge

the reviewer' concerns regarding the reported high prediction accuracy (e.g.,  $R^2 > 0.999$ , MRE < 2%) in the simulation results. While these test cases are critical for verifying model behavior and enabling comparative analysis, we agree that they cannot fully represent the complexities of real-world environments. We have already begun incorporating such tests into our ongoing work and have outlined this as a key direction for future development.

Lastly, we wish to clarify that although our study does not involve algorithmic invention per se, its primary contribution lies in application innovation and modular adaptability. We present a framework that not only achieves high performance under controlled conditions but is also sufficiently flexible to be extended to various field-scale groundwater inverse problems, including both point and non-point source contamination scenarios. Such an integrated and application-oriented modeling approach—particularly one that is computationally efficient and compatible with limited field data—is of direct relevance to both environmental practitioners and researchers. We hope these clarifications and revisions more accurately convey the intention, applicability, and potential of our work. The modified content is highlighted in red in the section 6.3. Please refer to lines 543-577 for details. Thank you again for your patient guidance and suggestions!

**To Reviewer 4:**

**General comments**

Good modelling research in the field of subsurface hydrology. Please, see my comments to fix the existing minor issues.

**Specific comments:**

- 1. Line 64. "Hydrogeological conditions". Insert recent papers on high-resolution datasets for determanation of hydrogeological conditions at contamianted sites.
- Maliva, R. G., Herrmann, R., Coulibaly, K., & Guo, W. (2015). Advanced aquifer characterization for optimization of managed aquifer recharge. Environmental Earth Sciences, 73, 7759-7767.
- Medici, G., Munn, J. D., & Parker, B. L. (2024). Delineating aquitard characteristics within a Silurian dolostone aquifer using high-density hydraulic head and fracture datasets. Hydrogeology Journal, 32, 1663-1691.

[Authors' response] Thank you for your suggestions. We have incorporated the relevant research literature you provided into the revised manuscript, which pertains to the application of high-resolution data in identifying hydrogeological conditions at contaminated sites, thereby enhancing the contextual explanation of the relevant content. The modified content is highlighted in red in the text. Please refer to line 65, lines 721-723 and lines 726-728 for details. Once again, we sincerely appreciate your thoughtful guidance and valuable suggestions!

**2. Line 151. MODFLOW, which version?**

[Authors' response] Thank you for your valuable feedback. The version of MODFLOW used in this study is MODFLOW-2005. The modified content is highlighted in red in the text. Please refer to line 161 for details. Thank you again for your careful guidance and valuable suggestions!

**3. Line 282. Specify the type of aquifer in terms of lithology.**

[Authors' response] Thank you for the comment. In the revised manuscript, we clarified the type of aquifer based on lithology. The aquifer consists of loose sediments, mainly well-sorted coarse sand and gravel, providing a clearer geological background. The modified content is highlighted in red in the text. Please refer to lines 302-304 for details. Thank you again for your careful guidance and valuable suggestions!

4. Line 302. Same here, specify the type of aquifer in terms of lithology.

[Authors' response] Thank you for the comment. In the revised manuscript, we clarified the type of aquifer based on lithology. The aquifer consists of loose sediments, mainly well-sorted coarse sand and gravel, providing a clearer geological background. The modified content is highlighted in red in the text. Please refer to lines 329-330 for details. Thank you again for your careful guidance and valuable suggestions!

5. Lines 340-341. "Mean relative error". I suggest Mean Absolute Relative Error because there is the modulus.

[Authors' response] Thank you for the suggestion. We agree with the reviewer that "Mean Absolute Relative Error" is the more accurate term, given the use of absolute values in the calculation. The modified content is highlighted in red in the text. Please refer to lines 359-360 and line 364 for details. Meanwhile, the corresponding part of the full text has also been corrected. Thank you again for your careful guidance and valuable suggestions!

**6. Line 521. Add a "take home message" for the researchers working in the field.**

[Authors' response] Thank you for the helpful suggestion. In the revised manuscript, we added a concise "take-home message" at the end of the Conclusion section to clearly summarize the key contributions and practical relevance of our study for researchers working on groundwater contamination source identification. The specific content is as follows: For researchers working in groundwater contamination source identification, this study underscores that method selection should not be guided solely by algorithmic novelty, but should be informed by the inherent complexity of the problem and the compatibility between the research question and the chosen approach. In groundwater contamination inversion, selecting a highly compatible method can substantially improve efficiency, while leveraging and organically integrating the strengths of different methods can greatly enhance robustness. This concept is equally applicable to a broader range of complex environmental inversion problems, offering valuable insights and practical potential. The modified content is highlighted in red in the text. Please refer to lines 632-640 for details. Thank you again for your patient guidance and

**suggestions!**

**Figures and tables**

1. Figure 5. Add the general flow direction with an arrow. Figure 5. Alternatively, divide the figures in two parts (A and B) adding the piezometric surfaces.

[Authors' response] Thank you for the suggestion. We added arrows to Figures 3 and 5 to indicate the approximate direction of groundwater flow and improve the clarity of the spatial background. Please refer to Figures 3 and 5 for details. Thank you again for your patient guidance and suggestions!

**2. Figure 6. I would add a spatial scale using a bar.**

[Authors' response] Thank you for the suggestion. We have added a spatial scale bar to Figure 4 and Figure 6 in the revised manuscript to enhance the interpretability of the spatial layout. Please refer to Figures 4 and 6 for details. Thank you again for your patient guidance and suggestions!

3. 9 tables are too many. Some of them can go in the Supplementary Material?

[Authors' response] Thank you very much for your thoughtful suggestions. We fully understand that nine tables may be burdensome for readers of the main text. Based on your feedback, we have moved Tables 2, 3, 5, and 9 to the supplementary materials and added references to the supplementary materials in the main text to ensure that readers can easily access them. The remaining tables, after careful consideration, contain core

information that is indispensable for supporting the research conclusions, and thus have been retained. Thank you again for your patient guidance and suggestions!

Thank you so much for your carefully review and good suggestions that make our paper quality improve. Best wishes for you and your whole family members!

Best wishes for you!

Sincerely

Chengming Luo, Xihua Wang, Y. Jun Xu, Qinya Lv, Xuming Ji, Boyang Mao, Shunqing Jia, Zejun Liu, Yanxin Rong, Yan Dai

August 3rd 2025

---

## Author Response (AR2)

**Responses to Editor and Reviewer:**

[Authors' response] First of all, we sincerely thank the editors and reviewers for their support and taking the time to review our manuscript. Your good suggestions have increased our papers quality. And also thank the editors to spend more time on our paper in the submitting process, thank you very much!

**To Reviewer 2:**

The authors have addressed most of my concerns. However, one aspect remains unclear, which largely limits the novelty of this study. Although the authors have compared some of the results with previous studies and added discussion in the revised manuscript (Lines 462–480), it is still not very clear from these discussions that this study makes substantial progress or introduces new ideas compared to earlier studies. The discussion mainly highlights similarities and consistencies with prior work.

[Authors' response] We sincerely thank the reviewer for the constructive comment. We fully agree that clearly highlighting the substantial progress and the unique contributions of this study is essential for emphasizing its novelty. In the revised manuscript, we have strengthened the discussion section.

Previous studies related to GCI employed a variety of methods to conduct either single or simultaneous inversion characterization of pollution sources and to identify hydrogeological parameters of the model. Li et al. (2022) identified the number, location, and release history of pollution sources, while Li et al. (2008) focused on determining the hydraulic conductivities of a study site. Bai et al. (2022) utilized

inversion techniques to simultaneously characterize pollution sources and identify the hydraulic conductivities within their simulation models. While some studies have applied inversion to the boundary conditions of the simulation model (Jiao et al., 2019), fewer studies have simultaneously characterized pollution sources and identified both hydrogeological parameters and boundary conditions of the model. Source information, model hydrogeological parameters, and boundary conditions are all critical components of groundwater contamination simulation models. Inaccuracies in any of these components can affect the overall results of inversion, making it essential to identify all components simultaneously. Therefore, in the PSC case of this study, the release history of the pollutant source, the hydraulic conductivity of the model, and the specific head boundary values were simultaneously identified. This simultaneous identification of multiple key parameters enhances the reliability and effectiveness of decision support systems.

One of the main methodological motivations of this study is the integration of the BPNN surrogate model with the AHA for GCI. This choice is grounded both in the inherent characteristics of GCI problems and in the complementary mechanisms of the two methods. GCI is a typical high-dimensional, nonlinear, and ill-posed inverse problem. The mapping from observed contaminant concentrations to source characteristics and hydrogeological parameters is often multimodal and nonconvex. In such cases, surrogate models such as BPNN can provide a fast and flexible approximation to computationally demanding groundwater simulations, but their use inevitably introduces approximation errors into the inversion objective function. These

errors may create local irregularities in the objective function landscape, which can mislead optimizers and cause premature convergence—particularly when the optimization algorithm lacks a mechanism to balance exploration and exploitation adaptively. AHA offers notable advantages in addressing these issues. Its bio-inspired mode-switching strategy alternates dynamically between diversified search and focused search. In the early stages of optimization, the broad and varied exploration capability helps to survey the global search space and reduces the risk of becoming trapped in spurious local optima caused by surrogate-induced noise. As the search proceeds, the algorithm adaptively shifts toward more intensive exploitation, concentrating computational effort on promising regions and thereby accelerating convergence. This dynamic adjustment is particularly important in GCI problems, where the optimal parameter region is often narrow and embedded within a complex and noisy search space. In addition, AHA's adaptive update mechanism adjusts search trajectories in response to population feedback, effectively mitigating the influence of local fluctuations in the surrogate-predicted objective function on the optimization process. This robustness to noisy or irregular fitness landscapes complements the BPNN's ability to generalize across diverse contamination scenarios. It is worth emphasizing that this integration is not a simple "algorithm replacement," but a targeted design choice based on the structural characteristics of the problem: BPNN provides broad adaptability to varying hydrogeological conditions, while AHA contributes resilience and fine-tuning capability when the optimization landscape is distorted by surrogate approximation errors. This synergy allows the proposed framework to maintain both high accuracy and strong robustness under different contamination scenarios and noise levels. More importantly, the underlying design principle—matching the characteristics of the surrogate model with the search dynamics of the optimization algorithm—has broader applicability to other environmental inversion problems. The modified content is highlighted in red. Please refer to lines 501-518 and lines 542-576 for details. Thank you again for your careful guidance and valuable suggestions!

**Minor comments:**

Lines 141-142 boundary conditions are not parameters. In addition, the innovations should be discussed in the context of other studies. Simply listing them in the introduction is not appropriate.

[Authors' response] We sincerely thank the reviewer for the constructive comment. We fully acknowledge that boundary conditions are not parameters, as this part of the original text has been revised. Additionally, in the discussion section, we further elaborate on the novelty of this study by contextualizing it within the background of other research. The modified content is highlighted in red. Please refer to lines 140-142, 501-518, and 542-576 for details. Thank you again for your careful guidance and valuable suggestions!

Lines 161-163 Here should cite the software developer, no?

[Authors' response] We appreciate the reviewer's valuable suggestion. We have cited the relevant software developers in the article. The additional content is highlighted in red. Please refer to lines 160-161, 674-675, and 821-822 for details. Thank you again for your careful guidance and valuable suggestions!

Line 585 Aquifer system is more accurate.

[Authors' response] We appreciate the reviewer's valuable suggestion. We have changed the term "groundwater system" in the article to "aquifer system." The modified content is highlighted in red. Please refer to line 585 for detail. Thank you again for your careful guidance and valuable suggestions!

**Remarks from the preceding review file validation:**

Please note that your reference list has not been compiled according to our standards.

Please consider adjusting your reference list with the next revision of your manuscript.

The manuscript preparation guidelines can be seen at: https://www.hydrology-and-earth-system-sciences.net/for\_authors/manuscript\_preparation.html.

[Authors' response] Thank you very much for your suggestion. We have formatted the references throughout the manuscript according to the preparation guidelines. The modified content is highlighted in red. Please refer to lines 656-822 for details. Thank you again for your careful guidance and valuable suggestions!

1. A "Short summary" system section contains scientific abbreviations. Please be aware that if you included scientific abbreviations (excluding chemical elements) without providing their full written explanations, you must write out them in full in your next

file upload request. However, do not forget that there is a limit to characters (not words!) for "Short summary": it must be < 500 characters.

[Authors' response] We have fully expanded the abbreviation "BP neural network" in the "Short Summary" to "back-propagation neural network." The total number of characters (including spaces) is 317, which is less than 500 characters. Thank you again for your careful guidance and valuable suggestions!

2. Please ensure that the colour schemes used in your maps and charts allow readers with colour vision deficiencies to correctly interpret your findings. Please check your figures using the Coblis – Color Blindness Simulator (https://www.color-blindness.com/coblis-color-blindness-simulator/) and revise the colour schemes accordingly with the next file upload request.

[Authors' response] Thank you very much for your suggestion. We have used Coblis—Color Blindness Simulator—to review the images in this article, ensuring that the color schemes employed in the figures enable readers with color vision deficiencies to accurately interpret our findings. Thank you again for your careful guidance and valuable suggestions!

3. Regarding the section "Author contributions", we kindly ask you to use rather initials of the authors instead of full names in this section with the next revision.

[Authors' response] Thank you very much for your suggestion. We have used the author's initials instead of their full names. The modified content is highlighted in red.

Please refer to lines 641-644 for details. Thank you again for your careful guidance and valuable suggestions!

4. Please add the figure captions directly beneath the respective figures.

[Authors' response] Thank you very much for your suggestion. We have added graphic titles directly below the corresponding figures. The modified content is highlighted in red. Please refer to lines 845-846, 848, 850, 852-853, 855, 857-858, 860-862, 865-866, and 868-869 for details. Thank you again for your careful guidance and valuable suggestions!

Thank you so much for your carefully review and good suggestions that make our paper quality improve. Best wishes for you and your whole family members!

Best wishes for you!

Sincerely

Chengming Luo, Xihua Wang, Y. Jun Xu, Qinya Lv, Xuming Ji, Boyang Mao, Shunqing Jia, Zejun Liu, Yanxin Rong, Yan Dai

August 28th 2025